# ID1 expressing macrophages support cancer cell stemness and limit CD8+ T cell infiltration in colorectal cancer

Shuang Shang [1,2,3,9], Chen Yang [1,2,3,9], Fei Chen [1,2,3], Ren-shen Xiang[4,5], Huan Zhang [1,2,3], Shu-yuan Dai [1,2,3], Jing Liu [1,2,3], Xiao-xi Lv[1,2,3], Cheng Zhang [1,2,3,6], Xiao-tong Liu [1,2,3], Qi Zhang[4,5], Shuai-bing Lu[4,5], Jia-wei Song [1,2,3], Jiao-jiao Yu[1,2,3], Ji-chao Zhou[1,3], Xiao-wei Zhang [1,2,3], Bing Cui [1,3], Ping-ping Li [1,3], Sheng-tao Zhu[7,8], Hai-zeng Zhang[4,5] ✉ & Fang Hua [1,2,3] ✉

Elimination of cancer stem cells (CSCs) and reinvigoration of antitumor immunity remain unmet challenges for cancer therapy. Tumor-associated macrophages (TAMs) constitute the prominant population of immune cells in tumor tissues, contributing to the formation of CSC niches and a suppressive immune microenvironment. Here, we report that high expression of inhibitor of differentiation 1 (ID1) in TAMs correlates with poor outcome in patients with colorectal cancer (CRC). ID1 expressing macrophages maintain cancer stemness and impede CD8+ T cell infiltration. Mechanistically, ID1 interacts with STAT1 to induce its cytoplasmic distribution and inhibits STAT1-mediated *SerpinB2* and *CCL4* transcription, two secretory factors responsible for cancer stemness inhibition and CD8+ T cell recruitment. Reducing ID1 expression ameliorates CRC progression and enhances tumor sensitivity to immunotherapy and chemotherapy. Collectively, our study highlights the pivotal role of ID1 in controlling the protumor phenotype of TAMs and paves the way for therapeutic targeting of ID1 in CRC.

Colorectal cancer (CRC) is the third-leading cause of cancer-related death worldwide. Local recurrence and distant metastasis after curative surgery are the major causes of mortality. Inherent and acquired resistance to chemotherapy or immunotherapy reduces the opportunity for patients to acquire long-term survival. Cancer initiation capacity and the suppressive tumor immune microenvironment are two pivotal determinants in driving tumorigenesis, progression, metastasis and therapy resistance[1,2]. Tumor-associated macrophages

[1]State Key Laboratory of Bioactive Substance and Function of Natural Medicines, Institute of Materia Medica, Chinese Academy of Medical Sciences & Peking Union Medical College, 100050 Beijing, P. R. China. [2]Beijing Key Laboratory of New Drug Mechanisms and Pharmacological Evaluation Study (BZ0150), Institute of Materia Medica, Chinese Academy of Medical Sciences & Peking Union Medical College, 100050 Beijing, P. R. China. [3]CAMS Key Laboratory of Molecular Mechanism and Target Discovery of Metabolic Disorder and Tumorigenesis, Institute of Materia Medica, Chinese Academy of Medical Sciences & Peking Union Medical College, 100050 Beijing, P. R. China. [4]Department of Colorectal Surgery, National Cancer Center/National Clinical Research Center for Cancer/Cancer Hospital, Chinese Academy of Medical Sciences and Peking Union Medical College, 100021 Beijing, P. R. China. [5]State Key Lab of Molecular Oncology, National Cancer Center/National Clinical Research Center for Cancer/Cancer Hospital, Chinese Academy of Medical Sciences and Peking Union Medical College, 100021 Beijing, P. R. China. [6]Department of Pharmacy, China–Japan Friendship Hospital, 100029 Beijing, P. R. China. [7]Beijing Digestive Diseases Center, Beijing Friendship Hospital, 100050 Beijing, P. R. China. [8]Beijing Key Laboratory for Precancerous Lesion of Digestive Diseases, Beijing Friendship Hospital, 100050 Beijing, P. R. China. [9]These authors contributed equally: Shuang Shang, Chen Yang. ✉e-mail: haizengzhang@163.com; huafang@imm.ac.cn

(TAMs) are the major components of immune cells in tumor micro-environment (TME)[3]. TAMs exhibit functional heterogeneity ranging from tumor-inhibiting to tumor-promoting phenotypes[4]. TAMs communicate with tumor cells or other immune cells by releasing a plethora of soluble mediators or expressing membrane docking molecules[5]. Numerous basic and clinical studies are ongoing to target TAMs for the treatment of tumor metastasis and enhancing sensitization to immunotherapy or chemotherapy[6]. A deeper understanding of the molecular mechanisms responsible for the phenotypic switch of TAMs is crucial for dissecting therapeutic resistance mechanisms and is helpful for developing rationalized combinatorial therapeutic approaches.

Transcriptome-based network analysis has identified that TAMs from cancer tissues are transcriptionally distinct from their respective tissue-resident macrophages and from their progenitor monocytes[5]. Transcriptomic analyses across many cancer types have revealed a prominent association between cancer stemness and immune signatures, potentially implying a biological interaction between such hallmark features of cancer. Indeed, factors, exosomes or metabolites secreted by cancer stem cells (CSCs) are known to recruit and induce polarization of TAMs[7–9]. Reciprocally, TAM-derived paracrine factors or physical interactions between TAMs and CSCs play prominent roles in supporting CSC stemness and the CSC niche[10,11]. Recently, some transcription factors (TFs) involved in maintaining the pluripotency and self-renewal characteristics of CSCs have been found to be highly expressed in TAMs and to determine the maintenance of the protumor phenotype of TAMs[12–14]. Targeting such transcription factors may be a potential therapeutic strategy for reversing cancer stemness and the suppressive TME simultaneously.

Inhibitor of differentiation 1 (ID1) is a member of the helix-loop-helix (HLH) transcriptional regulatory factors but has no DNA binding activity because of lacking the basic DNA binding domain. ID1 forms heterodimers with the basic HLH transcription factors to inhibit their DNA binding ability and transcriptional activity; therefore, it is also named as inhibitor of DNA-binding protein 1[15]. Although no tumor-associated mutations were observed in the *ID1* gene, it is still considered as a cancer-promoting factor. ID1 is highly expressed in a variety of human cancer types, such as colon, ovarian, pancreatic, esophageal cancers and glioblastoma[16–20]. High ID1 expression correlates with poor prognosis of cancer patients and increased tumor chemoresistance, angiogenesis, metastasis, and cancer stem cell properties. ID1 exerts its tumor-promoting effects by inhibiting the transcriptional function of several bHLH tumor suppressors, such as inhibiting the E2A-p21 axis to govern colon cancer-initiating cell self-renewal activity[16,21]. It also manifests tumor-accelerating effects via noncanonical mechanisms, including maintaining E2F1 protein stability to induce esophageal cancer chemoresistance and uncoupling TGFα-induced EMT from apoptosis in pancreatic cancer[17,18]. In addition to controlling tumor intrinsic properties, growing evidence indicates that ID1 is involved in the formation of an immunosuppressive microenvironment mediated by myeloid-derived suppressor cells (MDSCs) to promote melanoma progression[22]. However, the role of ID1 in shaping the immunosuppressive phenotype of TAMs, the most abundant immune population in the TME, remains an open question.

Here, we reveal that ID1 is highly expressed in CRC TAMs and is associated with poor clinical outcomes in CRC patients, supporting a role for ID1 in the maintenance of the protumor phenotype of TAMs. Furthermore, the enhanced expression of ID1 in TAMs exerts a pro-tumoral role by driving the acquisition of cancer stemness properties and excluding the CD8+ cytotoxic T lymphocyte infiltration into the TME via inhibition of signal transducer and activator of transcription 1 (STAT1)-mediated *SerpinB2* and *CCL4* transcription. Mechanistically, ID1 interacts with STAT1 to enhance the recruitment of chromosome region maintenance 1 (CRM1) to STAT1, which promotes STAT1 cytoplasmic distribution and inhibits its transcriptional activity.

Consequently, targeting ID1 inhibits CRC progression by synergizing with chemotherapy and immunotherapy. Our findings present the ID1-STAT1 regulatory cascade as a therapeutic target for switching TAMs toward an antitumorigenic phenotype for the treatment of CRC.

## Results

### Enhanced expression of ID1 in TAMs correlates with poor clinical outcomes for CRC patients

A transcriptome-based study revealed that TAMs from tumor tissues are transcriptionally distinct from monocytes and their respective tissue-resident macrophages[5]. To explore the differentially expressed oncogenic transcription factors in CRC TAMs, a gene microarray dataset (GSE80065) of peritoneal macrophages (PM) under the stimulation of CRC cell-derived conditioned medium (CM), was deeply mined. Among the 2830 upregulated genes (log2FC > 1.5, $P \le 0.05$) in TAMs (PM stimulated with CRC CM) compared to the unstimulated PMs (Ctrl), 80 genes were found to encode transcription factors, and 10 of them showed oncogenic properties in CRC (Fig. 1a). The protein abundance of the top 5 genes was validated in RAW 264.7 cells treated with CT26 or MC38-derived CM. c-Myc, Snai1, and Id1, but not Tcf4 and Id2, were found to be highly expressed in the colon cancer cell-derived CM-treated group compared to the control (Ctrl) group (Fig. 1b, c). As c-Myc and Snai1 have been demonstrated to be key players in alternative macrophage activation[12,23], we then focused on Id1 and its role in TAMs. Higher Id1 could be observed in TAMs infiltrated in CRC tissues from either the AOM/DSS-induced CRC model or *Apc*^Min spontaneous CRC model compared to the macrophages infiltrated in normal colon tissues (Fig. 1d–f). In addition, TAMs isolated from MC38-derived syngeneic orthotopic tumors exhibited higher Id1 mRNA and protein expression than PMs isolated from C57BL/6J mice (Fig. 1g, h). We also analyzed ID1 expression in human CRC TAMs by using tissue microarrays comprising 101 specimens from CRC patients. Higher ID1 expression was observed in CD68+ TAMs than in macrophages in the adjacent normal tissues (Fig. 1i, j). We also evaluated the potential kinetic alteration of ID1 expression in TAMs during CRC development. ID1 is highly expressed in TAMs of CRC patients with lymph node metastasis and positively correlated with CRC histologic grades and advanced TNM stages (Supplementary Table 1). Moreover, higher ID1 expression in CD68+ TAMs was associated with poor prognosis in patients with CRC (Fig. 1k). Collectively, our results indicate that ID1 is ectopically expressed highly in CRC TAMs, especially CRC with advanced stages, predicting poor clinical outcome.

### ID1 expressing TAMs promote CRC growth and metastasis

To determine whether ID1 directly regulates the macrophage phenotype, we established myeloid-specific *Id1*-deficient mice (referred to as *Id1*^Lyz-KO) by crossing mice containing floxed *Id1* exon (referred to as *Id1*^f/f) with mice expressing Cre recombinase under the control of the myeloid-specific gene promoter (referred to as *Lyz2*^tm1(cre)Ifo/J). Murine MC38 colon cancer cells were subcutaneously (s.c.) inoculated into *Id1*^f/f and *Id1*^Lyz-KO mice (Supplementary Fig. 1a). MC38 tumor-bearing *Id1*^Lyz-KO mice exhibited reduced tumor growth compared with *Id1*^f/f mice (Supplementary Fig. 1b–d), which is consistent with the data from the systemic *Id1*^KO mice[24]. The data indicated that Id1-expressing myeloid cells promote CRC progression. We further investigated whether ID1 ablation in macrophages inhibited tumor progression using two TAM adoptive transfer models. First, bone marrow-derived macrophages (BMDMs) were stimulated in vitro with basal medium presenting anti-inflammatory conditions (interleukin-4 (IL-4) and macrophage colony-stimulating factor (m-CSF)) to induce a TAM phenotype[25]. The aforementioned TAMs were mixed with MC38 cells and adoptively transferred into C57BL/6J recipient mice (Supplementary Fig. 1e). *Id1*^f/f-derived TAMs accelerated the tumor growth of MC38 cells, whereas depletion of *Id1* attenuated the tumor-promoting role of TAMs (Supplementary Fig. 1f–h). Second, TAMs were isolated from

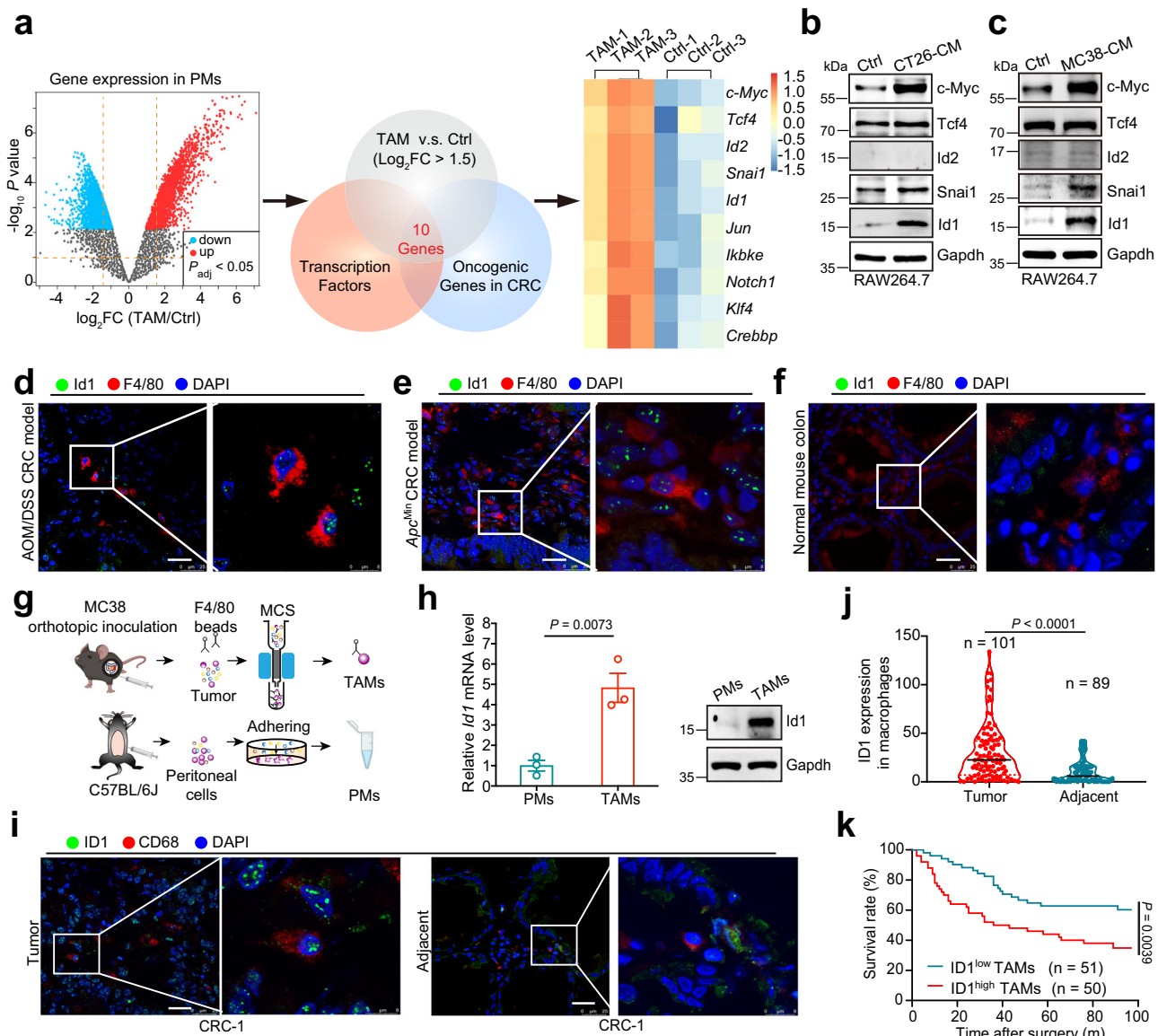

**Fig. 1 | Enhanced expression of ID1 in TAMs correlates with poor clinical outcome for CRC patients. a** Schematic diagram for screening the upregulated oncogenic transcription factors (Log$_2$FC > 1.5) in peritoneal macrophages (PMs) stimulated with CT26 cell-derived conditional medium (CM) compared to the untreated PMs (left) and the screened 10 upregulated genes (right) by using GEO dataset of GSE80065. **b, c** Immunoblots of indicated proteins in RAW264.7 cells treated with CT26-derived CM (**b**) or MC38-derived CM (**c**), $n$ = 3 biologically independent samples. **d–f** Representative multiple immunohistochemistry (mIHC) images of Id1 and F4/80 in tumor tissues of AOM/DSS-induced CRC model (**d**), *Apc*$^{Min}$ spontaneous CRC model (**e**) or in the normal colon tissues of C57BL/6J mouse (**f**). Scale bar, 25 μm, $n$ = 3 biologically independent samples. **g, h** The

strategy of isolating TAMs from orthotopic MC38 tumor-bearing mice or PMs from normal C57BL/6 J mice (**g**). The mRNA (**h**, left) and protein abundance (**h**, right) of Id1 in TAMs and PMs were detected, $n$ = 3 mice per group, Student's *t*-test. **i, j** Representative mIHC staining of ID1 and CD68 in tumor and adjacent normal tissues from CRC patients (**i**), and the related statistical data of ID1 expression within CD68$^+$ cells (**j**), Mann–Whitney *U* test. Scale bar, 25 μm. **k** Kaplan–Meier plot of overall survival of CRC patients stratified by ID1 expression level within CD68$^+$ cells, Log-rank test. Unless specified otherwise, the data are presented as means ± SEM. **g** is created with BioRender.com. Source data are provided as a Source Data file.

MC38-derived tumors in *Id1*$^{f/f}$ and *Id1*$^{Lyz-KO}$ mice with magnetic cell sorting, mixed with MC38 cells, and adoptively transferred into C57BL/6J recipient mice (Fig. 2a). Tumors containing *Id1*$^{Lyz-KO}$ TAMs grew much slower than those containing *Id1*$^{f/f}$ TAMs (Fig. 2b, c). The tumor nodules from the *Id1*$^{Lyz-KO}$ TAM group showed a lower ratio of Ki67$^+$ cells (Fig. 2d). These data indicate that depletion of *Id1* in TAMs inhibits CRC growth.

TAMs often engage in crosstalk with tumor cells or other immune cells in the TME by influencing the production of secretory factors[26]. MC38 tumor-bearing mice were intratumorally (i.t.) injected with CM derived from *Id1*$^{f/f}$ or *Id1*$^{Lyz-KO}$ TAMs (Supplementary Fig. 1i). Tumors injected with CM from *Id1*$^{Lyz-KO}$ TAMs grew slower than those injected

with CM from *Id1*$^{f/f}$ TAMs (Supplementary Fig. 1j–l). Similar results were observed in the CT26 s.c. model with intratumoral injection of TAM-derived CM (Supplementary Fig. 1m–p). To investigate whether ID1 in macrophages participates in the control of tumor metastasis, three CRC metastasis models were established. In the spleen–liver metastasis model, luciferase-labeled CT26 cells were intra-splenic injected together with intraperitoneal injection (i.p.) of TAM-derived CM (Fig. 2e). Mice injected with CM-derived from *Id1*$^{f/f}$ TAMs showed an enhanced bioluminescent signal, liver/body weight ratio, tumor diameter and number of hepatic lobes with metastatic tumor nodules, which were reversed when *Id1* was ablated in TAMs (Fig. 2f–k). In the CT26 orthotopic liver metastasis model, highly metastatic CT26 cells

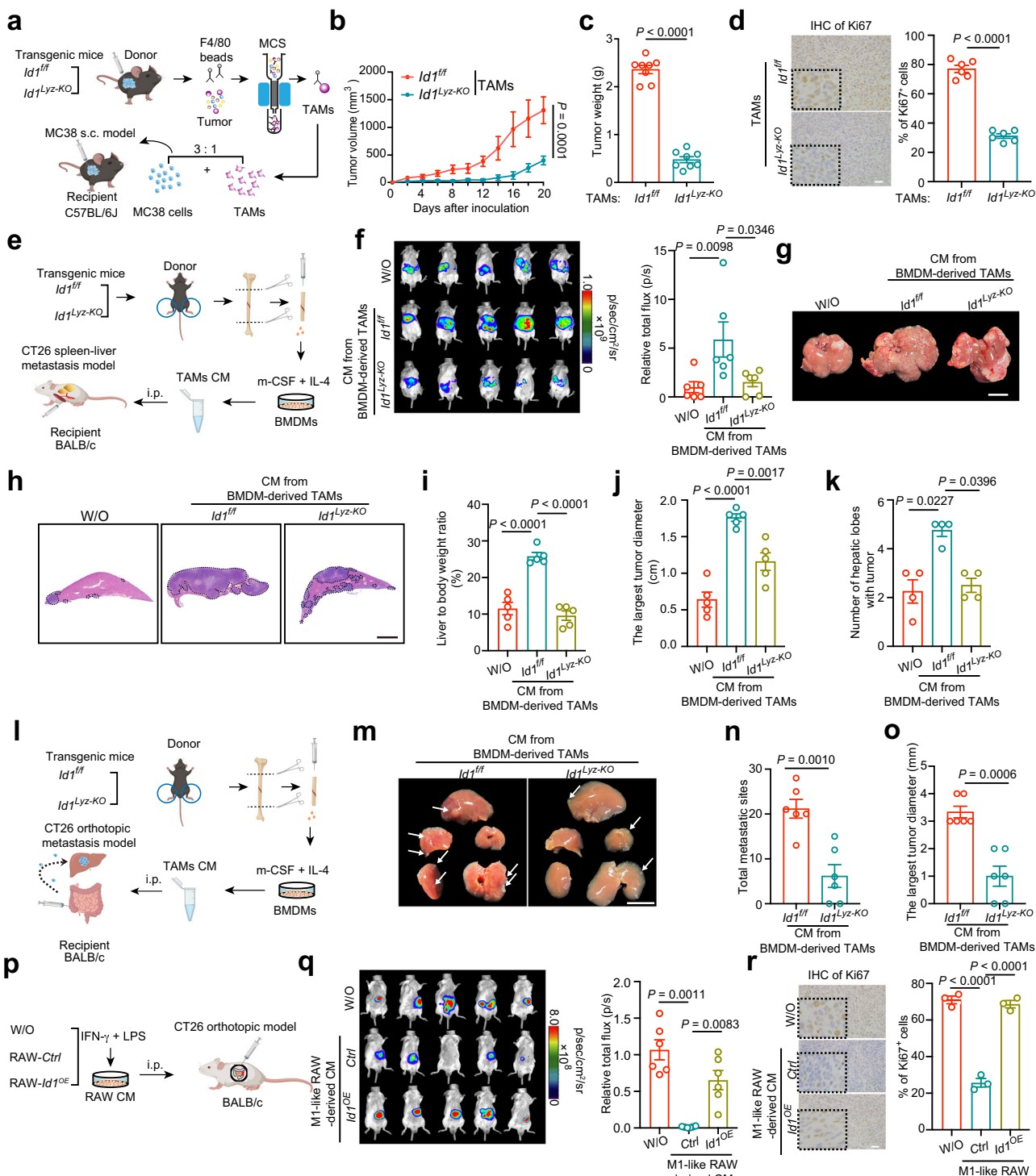

**Fig. 2 | ID1 expressing TAMs promote CRC growth and metastasis. a** Schematic diagram for establishing the MC38 and TAMs mixture s.c. model. **b, c** Tumor volumes (**b**) and tumor weight (**c**) of groups presented in (**a**), $n = 8$ mice per group, Mann–Whitney $U$ test is used in (**b**), and Welch's test is used in (**c**). **d** Representatives of Ki67 immunohistochemical staining in the groups presented in (**a**), $n = 6$ mice per group, Student's $t$-test. Scale bar, 25 μm. **e** Schematic diagram for establishing the CT26 spleen–liver metastasis model. **f** Representative bioluminescence images and statistical data of bioluminescence signal of groups presented in (**e**), $n = 6$ mice per group, Kruskal–Wallis test. W/O: without. **g, h** Representative gross liver images (**g**) and H&E staining (**h**) of the liver lobes for groups as presented in (**e**). Scale bar, 1 cm. **i** The ratio of liver weight to body weight of groups presented in (**e**), $n = 5$ mice per group, one-way ANOVA test. **j** The largest diameter of metastatic tumor of groups as presented in (**e**), $n = 5$ mice per group,

one-way ANOVA test. **k** The number of hepatic lobes with metastatic tumor nodules of groups presented in (**e**), $n = 4$ mice per group, Kruskal–Wallis test. **l** Schematic diagram for establishing the CT26 orthotopic liver metastasis model. **m** Representative gross liver images of groups presented in (**l**). Scale bar, 1 cm. **n** Total metastatic sites of groups presented in (**l**), $n = 6$ mice per group, Student's $t$-test. **o** The largest tumor diameter of groups presented in (**l**), $n = 6$ mice per group, Student's $t$-test. **p** Schematic diagram for establishing the CT26 orthotopic model. **q** Representative bioluminescence images and statistical data of bioluminescence signal of groups presented in (**p**), $n = 6$ mice per group, one-way ANOVA test. **r** Representative immunohistochemical Ki67 staining of groups presented in (**p**). Scale bar, 25 μm. $n = 3$ mice per group, one-way ANOVA test. **a, e, l**, and **p** are created with BioRender.com. Source data are provided as a Source Data file.

were injected into the subserous layer of the cecal wall together with i.p. injection of TAM-derived CM (Fig. 2l). Fewer metastatic sites and smaller metastatic tumor diameter were observed in the group treated with CM-derived from $Id1^{Lyz-KO}$ TAMs (Fig. 2m–o). In the CRC lung metastasis model, luciferase-labeled CT26 cells were injected via the tail vein together with i.p. injection of TAM-derived CM. Less lung metastatic region and lower lung/body weight ratio were observed in the group treated with CM-derived from $Id1^{Lyz-KO}$ TAMs (Supplementary Fig. 1q–s). These data suggested that ID1-expressing TAMs promote CRC tumor growth and metastasis by affecting the secreted components of TAMs. It has been reported that proinflammatory stimuli induce an antitumorigenic (M1-like) phenotype of macrophages[27]. RAW 264.7 cells (shown as RAW-*Ctrl*) and RAW 264.7 cells with *Id1* overexpression (shown as RAW-$Id1^{OE}$) were stimulated with IFN-γ and LPS to induce the tumor inhibition phenotype (Fig. 2p). We found that the CM from RAW-*Ctrl* cells alleviated tumor cell growth and reduced the proportion of Ki67+ cells, while *Id1* ectopic expression reversed these inhibitory effects (Fig. 2q, r). These data suggested that high Id1 expression impairs the antitumor phenotype of classically activated macrophages.

### ID1 expressing TAMs promotes CRC development partially through excluding CD8+ T cell recruitment

TAMs usually have direct immunosuppressive functions by preventing tumor cells from being attacked by T cells. To evaluate whether ID1 expressing TAMs play immune suppressive functions, a mixture of MC38 cells and TAMs, as presented in Fig. 2a, was inoculated into immunodeficient BALB/c nude mice instead of immunocompetent C57BL/6J mice (Supplementary Fig. 2a). MC38 cells mixed with $Id1^{Lyz-KO}$ TAMs showed reduced tumor growth compared with those mixed with $Id1^{f/f}$ TAMs (Supplementary Fig. 2b–d). The tumor inhibition rate in BALB/c nude mice was approximately 50%, whereas that in C57BL/6J mice (as presented in Fig. 2a) was more than 80% (Fig. 3a), indicating that ID1 expressing TAMs may destroy the T cell-mediated antitumor immune response. Deletion of *Id1* in TAMs increased CD8+ T cell infiltration in MC38-derived tumors from immunocompetent C57BL/6J mice (Fig. 3b, c). In addition, depletion of *Id1* in TAMs increased the expression of interferon-γ (IFN-γ) and Granzyme B in tumor-infiltrating CD8+ T cells (Fig. 3d). We also evaluated the involvement of ID1 in the antitumor immune response using a spleen–liver metastatic model in BALB/c nude mice (Supplementary Fig. 2e). Treatment with CM from $Id1^{Lyz-KO}$ TAMs alleviated the bioluminescent signal, liver/body weight ratio, and tumor diameter in BALB/c nude mice compared to CM from $Id1^{f/f}$ TAMs (Supplementary Fig. 2f–j). However, the tumor inhibition rate resulting from *Id1* depletion in TAMs decreased from ~70% in immunocompetent mice to ~40% in immunodeficient mice (Fig. 3e), confirming ID1 expressing TAMs in the regulation of T cell-mediated antitumor immunity. Treatment with CM from $Id1^{f/f}$ TAMs decreased CD8+ T cell infiltration in the intratumoral regions of liver or lung metastases (as presented in Fig. 2e, l and Supplementary Fig. 1q), which was reversed by knocking out of *Id1* in the TAMs (Fig. 3f–h). In addition, treatment with CM from M1-like RAW 264.7 cells in an orthotopic colorectal cancer model (as presented in Fig. 2p) increased CD8+ T cell infiltration in the tumor, which was abrogated by *Id1* overexpression (Fig. 3i). To validate the involvement of CD8+ T cells in the tumor-promoting role of ID1 expressing TAMs, we depleted CD8+ T cells using systemic administration of an anti-CD8β antibody in a CT26 s.c. model (Fig. 3j). Treatment with CM from $Id1^{Lyz-KO}$ TAMs reduced tumor growth compared to the $Id1^{f/f}$ group, which was partially reversed by CD8+ T cells depletion (Fig. 3k–m), with a tumor inhibition rate ranging from ~85% (IgG treatment group) to 35% (CD8 depleting antibody group) (Fig. 3n). These data suggest that the tumor-promoting role of ID1-expressing TAMs is CD8+ T cell-dependent.

Lack of intratumoral T cell infiltration is associated with reduced T cell proliferation or migration. $Id1^{OE}$ in M1-like RAW 264.7 cells

inhibited CD8+ T cell migration, whereas $Id1^{Lyz-KO}$ in TAMs enhanced CD8+ T cell migration (Fig. 3o, p). We also evaluated CM from macrophages with Id1 manipulation on T cell proliferation and killing activity. Labeling CD8+ T cells with carboxyfluorescein succinimidyl ester (CFSE) fluorescent dye, we found that overexpression of *Id1* in M1-like RAW 264.7 cells or depletion of *Id1* in TAMs had no effects on CD8+ T cell proliferation (Supplementary Fig. 2k). Moreover, no difference was observed between $Id1^{f/f}$ and $Id1^{Lyz-KO}$ groups in influencing OVA-specific CD8+ cytotoxic OT-1 T cells killing activity (Supplementary Fig. 2l). We also confirmed our observations in human primary cells co-culture system. When co-culturing with CD8+ T cells isolated from CRC patients' peripheral blood mononuclear cells (PBMCs), *ID1* depletion in TAMs isolated from CRC patients' tumor tissues showed enhanced CD8+ T cell migration rate in comparison with the *Ctrl* group (Fig. 3q). ID1 expressing MDSCs were reported to promote tumor growth[22], we further explored whether they also exerted the tumor-promoting role through influencing CD8+ T cell infiltration. Cd11b+ Gr1+ MDSCs isolated from MC38-derived tumors in $Id1^{f/f}$ and $Id1^{Lyz-KO}$ mice were cocultured with CD8+ T cells. We found that depletion of *Id1* in MDSCs had no effect on CD8+ T cells migration rate (Supplementary Fig. 2m). These data suggested that ID1 expressing TAMs but not MDSCs hamper CD8+ T cell recruitment and promote the evasion of tumor cells from immune elimination.

### ID1-expressing TAMs are essential for the maintenance of CRC stemness traits via activating FAK-YAP signaling

Knockout of *Id1* in TAMs showed an inhibitory effect on tumor growth and metastasis in immunodeficient mice (as presented in Supplementary Fig. 2a–j), suggesting that ID1 expressing TAMs could affect tumor intrinsic properties. CSCs are considered the origin of tumorigenesis, metastasis, drug resistance, and relapse. We evaluated ID1 in TAMs with respect to cancer stemness trait maintenance. CD45-negative but epithelial cell adhesion molecule (Epcam) positive (CD45− Epcam+) tumor cells were isolated from tumor nodules in s.c. MC38 model (as presented in Supplementary Fig. 2a) or from metastases in the CT26 orthotopic liver/lung metastasis model (as presented in Fig. 2l; Supplementary Fig. 1q). Tumor cells isolated from $Id1^{Lyz-KO}$ group as mentioned above expressed lower CD44 and Lgr5 (CSC marker) than those from $Id1^{f/f}$ group (Fig. 4a, b; Supplementary Fig. 3a). Depletion of *Id1* in TAMs decreased the tumor-initiating cell frequency to nearly 1/3 (from 1/979 to 1/2,378) (Fig. 4c). These data demonstrated that ID1 expressing TAMs support the stemness traits of CRC cells. TAMs can foster CSC phenotypes through soluble mediators[28]. HCT116 cells cocultured with CRC patient's tumor tissues derived $ID1^{KD}$ TAMs expressed lower levels of CD44 and LGR5 than those cocultured with *Ctrl* TAMs (Fig. 4d). MC38 cells cocultured with $Id1^{Lyz-KO}$ TAMs expressed lower levels of CD44 and Aldh than those cocultured with $Id1^{f/f}$ TAMs (Fig. 4e). CT26 cells cocultured with *Id1*-overexpressing M1-like RAW 264.7 cells, and HCT116 cells cocultured with *ID1*-overexpressing M1-like THP-1 cells showed higher CD44 and ALDH expression than those cocultured with *Ctrl* cells (Supplementary Fig. 3b, c). In addition, CM from $Id1^{KD}$ or $Id1^{Lyz-KO}$ TAMs was inhibited, while CM from $Id1^{OE}$ RAW 264.7 cells or $ID1^{OE}$ THP-1 cells promoted, the tumor sphere formation ability (Fig. 4f; Supplementary Fig. 3d–f) and invasiveness of CRC cells (Supplementary Fig. 3g, h). These data suggested that ID1 in TAMs increases CRC cell stemness traits by changing the secreted components of TAMs.

Transcriptome sequencing was employed to assess differentially expressed genes (DEGs) in cancer cells isolated from s.c. tumor nodules which were implanted by mixing MC38 cells and $Id1^{f/f}$/$Id1^{Lyz-KO}$ TAMs (Fig. 4g). Gene set enrichment analysis (GSEA) suggested that *Id1* deletion in TAMs resulted in the inhibition of focal adhesion kinase (FAK) and Yes-associated protein (YAP) signaling pathways but not other tumor stemness associated signaling such as Notch, Wnt/β-catenin and Sonic hedgehog (SHH) pathway (Fig. 4h, i; Supplementary

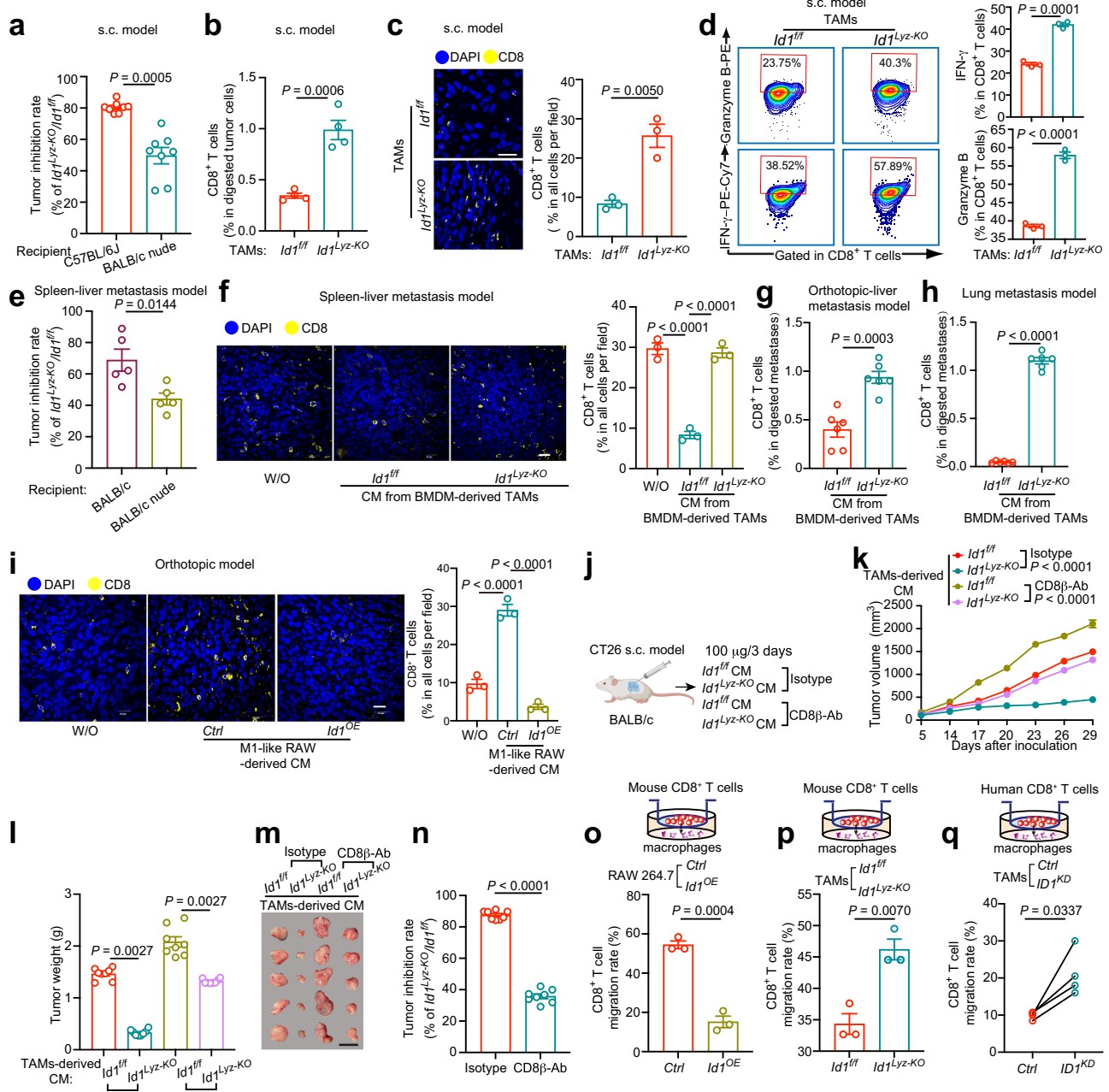

**Fig. 3 | ID1-expressing TAMs promote CRC progression partially through inhibiting CD8+ T cell recruitment. a** Tumor inhibition rate of MC38 and TAMs mixture s.c. model in the indicated groups, *n* = 8 mice per group, Welch's test. **b**, **c** Percentage of CD8+ T cells (**b**), representative mIHC images, and statistical data of CD8+ T cells (**c**) infiltrated in the tumor tissues as presented in Fig. 2a, *n* = 4 (**b**) or 3 (**c**) mice per group, Student's *t*-test, Scale bar, 25 μm. **d** Representative density dot plots and the statistical data for IFN-γ+ and Granzyme B+ CD8+ T cells in the tumor tissues are presented in Fig. 2a. *n* = 3 mice per group, Student's *t*-test. **e** Tumor inhibition rate of CT26 spleen-liver metastasis model with i.p. injection of CM from BMDM-derived TAMs in the indicated groups, *n* = 5 mice per group, Student's *t*-test. **f** Representative mIHC images and statistical data of CD8+ T cells in liver metastatic tumor tissues presented in Fig. 2e, *n* = 3 mice per group, one-way ANOVA test. Scale bar, 20 μm. **g**, **h** Percentage of CD8+ T cells infiltrated in the tumor tissues from

recipient mice presented in Fig. 2l (**g**) and Supplementary Fig. 1q (**h**), *n* = 6 mice per group, Student's *t*-test. **i** Representative mIHC images and statistical data of CD8+ T cells in tumors presented in Fig. 2p, *n* = 3 mice per group, one-way ANOVA test. Scale bar, 20 μm. **j** Schematic diagram for the deletion of CD8+ T cells in tumor models. **k**–**m** Tumor volumes (**k**), tumor weight (**l**), and representative tumor images (**m**) of groups presented in (**j**), *n* = 8 mice per group, Kruskal–Wallis test. Scale bar, 1 cm. **n** Tumor inhibition rate of indicated groups, *n* = 8 mice per group, Student's *t*-test. **o**, **p** Relative migration of CD8+ T cells cocultured with different groups of RAW264.7 cells (**o**) or TAMs (**p**), *n* = 3 biologically independent samples, Welch's test. **q** Relative migration of human CD8+ T cells cocultured with different groups of TAMs, *n* = 4 patients, paired *t*-test. **j** is created with BioRender.com. Source data are provided as a Source Data file.

Fig. 3i[29,30]. Consistent with this data, tumor cells isolated from tumor nodules mixed with BMDM-derived TAMs (as presented in Supplementary Fig. 1e) from *Id1f/f* mice exhibited a higher abundance of p-Fak and Yap compared to the nonmixed cells, which were reversed by depleting of *Id1* in TAMs (Fig. 4j). We also detected the FAK-YAP

signaling changes in the MC38 s.c. model (as presented in Fig. 2a), the spleen-liver metastasis model (as presented in Fig. 2e), the orthotopic model (as presented in Fig. 2p), as well as an in vitro cell non-contact coculture system. TAMs with *Id1* deletion decreased, while M1-like RAW 264.7 cells with *Id1* overexpression increased the abundance of

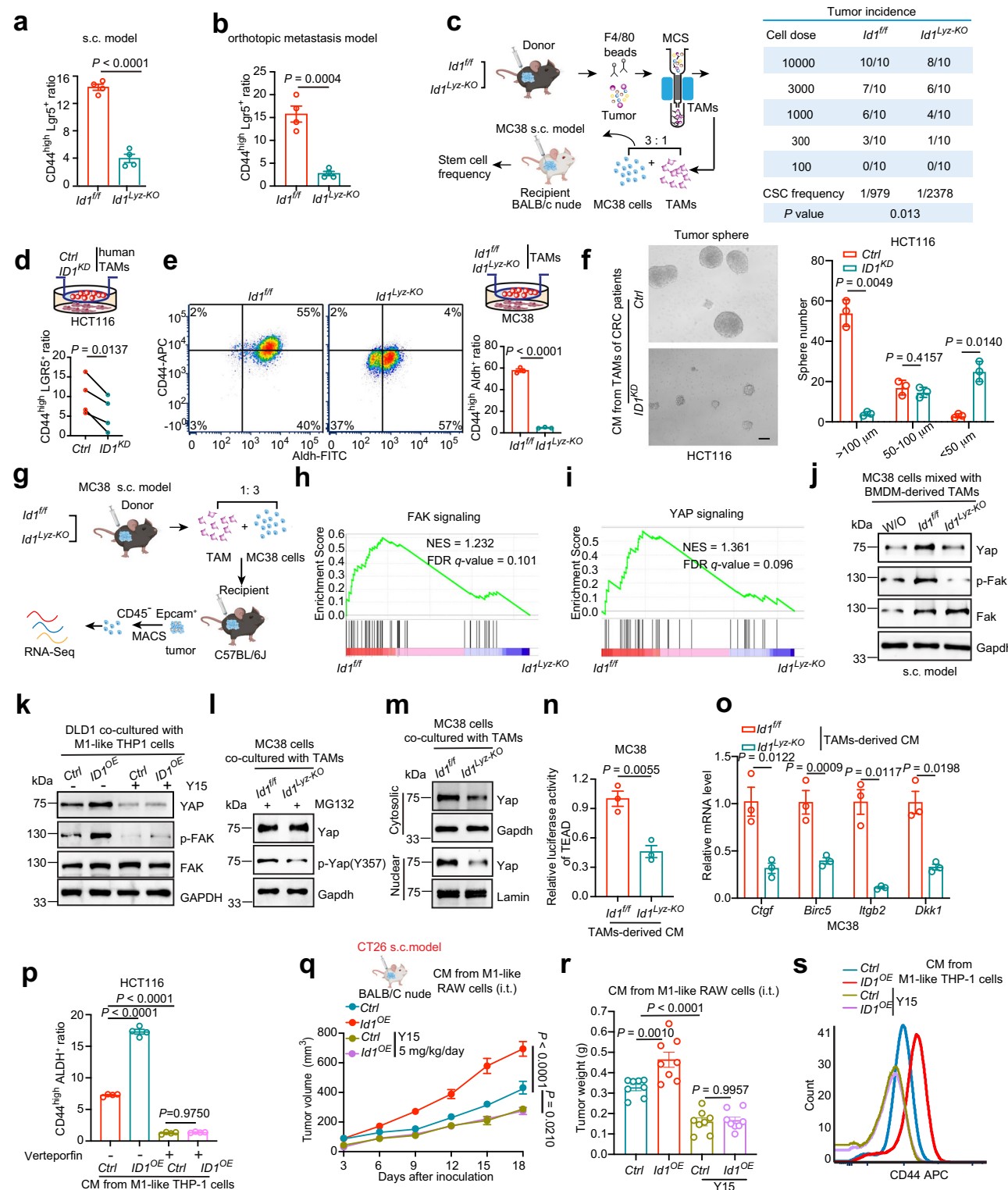

p-Fak and Yap (Supplementary Fig. 3j–m). Y15, a specific inhibitor of FAK, could reverse the upregulation role of *ID1* overexpression in M1-like THP1 cells on YAP protein expression, indicating that the ID1 induces YAP activation via a FAK-dependent manner (Fig. 4k). FAK maintains the tumor stemness traits via phosphorylating YAP at Y357 to enhance its protein stability, nuclear translocation, and its transcriptional activity[31]. YAP forms a transcriptional complex with TEAD after nuclear translocation, which is a key step in maintaining tumor stemness traits[30]. *ID1* depletion downregulated while its overexpression enhanced YAP protein stability, YAP (Y357)

phosphorylation, nuclear translocation, the luciferase activity of TEAD, and the transcription of downstream target genes (Fig. 4l–o and Supplementary Fig. 3n–q). Verteporfin, a suppressor of YAP–TEAD complex, largely reversed the CRC tumor stemness-promoting effect of CM from *ID1* overexpressing M1-like THP-1 cells (Fig. 4p). These data suggested that the secretory components from ID1 expressing TAMs are crucial for maintaining the FAK-YAP activation in tumor cells. Y15 was further used to verify whether ID1 expressing TAMs maintain cancer stemness traits through activation of FAK signaling. In the CT26 s.c. BALB/c nude mouse model, Y15 alleviated tumor growth as

**Fig. 4 | ID1-expressing TAMs are essential for the maintenance of CRC stemness traits through activating FAK-YAP signaling. a, b** Flow cytometry analysis of CD44$^{high}$ Lgr5$^+$ cell ratio in CD45$^-$ Epcam$^+$ tumor cells isolated from MC38-derived tumor nodules (**a**) in Supplementary Fig. 2a or CT26-derived liver metastases (**b**) presented in Fig. 2l, $n = 4$ biologically independent samples, Student's $t$-test. **c** Schematic diagram for determining the tumor initiation capacity and the statistical data of CSC frequency. **d** Flow cytometry analysis of CD44$^{high}$ LGR5$^+$ cell ratio in HCT116 cells cocultured with *Ctrl* or *ID1$^{KD}$* TAMs, $n = 4$ patients, paired $t$-test. **e** Flow cytometry analysis of CD44$^{high}$ Aldh$^+$ cell ratio in different groups of MC38 cells, $n = 3$ biologically independent samples, Student's $t$-test. **f** Images and quantification of HCT116 tumor spheres with different treatments, $n = 3$ biologically independent samples, Welch's test. **g** Schematic diagram for the transcriptome sequencing. **h, i** GSEA analysis on differentially expressed genes between CD45$^-$ Epcam$^+$ tumor cells of indicated groups presented in (**g**) with a predefined gene set of FAK (**h**) and YAP signaling (**i**). $n = 3$ biologically independent samples per group in the RNA-seq data. **j** Immunoblots of indicated proteins in different groups of CD45$^-$ Epcam$^+$ tumor cells presented in Supplementary Fig. 1e, $n = 3$ biologically independent samples. **k** Immunoblots of indicated proteins in DLD-1 cells cocultured with different THP-1 cells with or without Y15 treatment, $n = 3$ biologically independent samples. **l** Immunoblots of indicated proteins in MC38 cells with different treatments, $n = 3$ biologically independent samples. **m** Immunoblots of cytosolic and nuclear Yap in MC38 cells cocultured with different TAMs, $n = 3$ biologically independent samples. **n** Relative luciferase activity of TEAD in MC38 cells cocultured with different TAMs, $n = 3$ biologically independent samples, Student's $t$-test. **o** Relative mRNA expression of Yap downstream genes in different groups of MC38 cells, $n = 3$ biologically independent samples, Student's $t$-test. **p** Effects of Verteporfin on CD44$^{high}$ ALDH$^+$ ratio of HCT116 cells cocultured with different groups of THP-1 cells, $n = 3$ biologically independent samples. **q** Schematic diagram for tumor model (**q**, up) and tumor volumes (**q** down) of the indicated groups, one-way ANOVA test, $n = 12, 10, 10,$ and 8 mice in the groups of *Ctrl, Id1$^{OE}$, Ctrl*-Y15, *Id1$^{OE}$*-Y15. **r** Schematic diagram for tumor weight of the indicated groups, $n = 8$ mice per group, one-way ANOVA test. **s** Effects of Y15 on CD44 expression of DLD-1 cells cocultured with different groups of THP-1 cells, $n = 3$ biologically independent samples. **c, g,** and **q** are created with BioRender.com. Source data are provided as a Source Data file.

expected and reversed the tumor-accelerating effect of CM from *Id1$^{OE}$* RAW 264.7 cells compared to the *Ctrl* cells (Fig. 4q, r). Notably, under Y15 treatment, CM from *Id1$^{OE}$* M1-like macrophages no longer enhanced tumor cell invasiveness, tumor sphere-forming ability, or CD44 protein abundance (Fig. 4s; Supplementary Fig. 3r, s). These data suggested that ID1-expressing TAMs support CRC stemness by activating FAK-YAP signaling in cancer cells.

## ID1 in TAMs mediates tumor immune evasion and CRC stemness maintenance by inhibiting *CCL4* and *SerpinB2* transcription

Transcriptome sequencing of TAMs isolated from MC38-derived tumor tissues in *Id1$^{f/f}$* or *Id1$^{Lyz-KO}$* mice was performed to explore how Id1 shaped the tumor-promoting phenotype of TAMs (Fig. 5a). ID1 inhibits the DNA binding and transcriptional activation ability of bHLH proteins with which it interacts[32]. Our above data indicated that the secretory components regulated by ID1 are crucial for TAMs in maintaining cancer stemness and promoting cancer immune invasion. Therefore, we focused on the upregulated genes encoding secretory proteins in *Id1$^{Lyz-KO}$* TAMs compared to *Id1$^{f/f}$* TAMs. In *Id1$^{Lyz-KO}$* TAMs, over 2000 genes were found to be increased when compared to *Id1$^{f/f}$* TAMs (Fig. 5b). From the immune response regulation aspect, 20 upregulated genes encoding chemokines with log$_2$FC > 1.5 were selected, among which *Ccl3, Ccl4, Cxcl9* and *Cxcl16* shared with both tumor suppressive roles and effects in promotion of T cell infiltration (Fig. 5c)[33–36]. Among the four chemokines, only *Ccl4* was found to exhibit lower transcription in *Id1$^{OE}$* RAW 264.7 cells than in *Ctrl* cells (Fig. 5d). According to the CRC GEO dataset, low *CCL4* correlates with poor outcome in patients with CRC (Supplementary Fig. 4a). Enzyme-linked immunosorbent assay (ELISA) validated that TAMs with *Id1* depletion, isolated from CRC patient tumor tissues, MC38-derived tumor tissues or induced from BMDMs, showed increased CCL4 protein abundance (Fig. 5e; Supplementary Fig. 4b, c). *ID1* overexpression in M1-like macrophages reduced CCL4 protein levels (Fig. 5f; Supplementary Fig. 4d). We then established *Ccl4* depleted TAMs to validate the role of Ccl4 in mediating Id1-regulated CD8$^+$ T cell recruitment (Supplementary Fig. 4e). *Id1* deletion in TAMs resulted in increased CD8$^+$ T cell migration, which could be reversed by knocking down of *Ccl4* (Fig. 5g). These data indicate that ID1 expressing TAMs lose the secretion of CCL4 and hinder CD8$^+$ T cell trafficking to tumor sites.

From the tumor intrinsic property regulation aspect, 12 upregulated genes encoding secretory tumor suppressive factors (exclusive of chemokines) with log$_2$FC > 1.5 were selected, among which only Serpin family B member 2 (SerpinB2) was reported to inhibit FAK signaling (Fig. 5h)[37]. Analyzing the CRC GEO dataset, we found that *SerpinB2* was positively correlated with satisfactory outcomes of CRC patients (Supplementary Fig. 4f). *ID1* depletion in TAMs increased

SerpinB2 protein abundance, while *ID1* overexpression decreased the abundance (Fig. 5i, j; Supplementary Fig. 4g–i). The inhibitory effect of *Id1* knockout in TAMs on CSC marker expression, tumor sphere formation ability, and tumor invasiveness could be reversed by depletion of Serpinb2 (*Serpinb2$^{KD}$*) (Supplementary Fig. 4j; Fig. 5k–m). In addition, silencing *Serpinb2* in TAMs enhanced the tumor invasiveness and tumor sphere-forming ability, which could be reversed by Y15 treatment (Supplementary Fig. 4k, l). These data indicate that ID1 inhibits the expression of SerpinB2 to activate FAK signaling and enhance cancer cell stemness.

To confirm whether ID1 played a tumor-promoting role by impairing SerpinB2 and CCL4 expression, the two genes were simultaneously depleted in *Id1$^{f/f}$* and *Id1$^{Lyz-KO}$* TAMs. The inhibitory effects of *Id1* deletion in TAMs on both tumor growth and liver metastasis were completely abrogated when *Ccl4* and *Serpinb2* were depleted simultaneously (Fig. 5n–p; Supplementary Fig. 4m). Either in the s.c. model or in the metastasis model, the enhancement of CD8$^+$ T cell infiltration and the inhibition of FAK-YAP signaling caused by *Id1* deletion in TAMs were both diminished when *Ccl4* and *Serpinb2* were knocked down simultaneously (Supplementary Fig. 4n–q). The evidence suggested that ID1 inhibits the transcription of *SerpinB2* and *CCL4* in macrophages to promote tumor immune evasion and augment the tumor-initiating capacity of CRC cells, ultimately resulting in aggravated tumor growth and metastasis.

## ID1 interacts with STAT1 to inhibit the transcription of *CCL4* and *SerpinB2*

As ID1 has no DNA binding ability and usually interacts with and inhibits the activity of TFs, coimmunoprecipitation coupled with mass spectrometry (Co-IP/MS) was performed to identify the potential Id1-interacting TFs in RAW 264.7 cells. In addition, 16 possible TFs responsible for *CCL4* and *SerpinB2* transcription were predicted using the hTFtarget tool[38]. STAT1 is the only one among the 16 TFs that interact with ID1 according to Co-IP/MS analysis (Fig. 6a). We further confirmed the direct interaction between STAT1 and ID1 in HEK 293T cells, RAW 264.7 cells, and CD68$^+$ TAMs in CRC patient tumor tissues by using Co-IP, proximity ligation assay (PLA) and confocal imaging (Fig. 6b-e). These data indicated that ID1 might interact with STAT1 to inhibit STAT1-mediated *CCL4* and *SerpinB2* transcription. A chromatin immunoprecipitation (ChIP) assay was used to verify whether *Serpinb2* and *Ccl4* are the target genes of Stat1 in RAW 264.7 cells. Stat1 bound to the promoter of *Ccl4* at −3000 to −2701 bp, −2760 to −2461 bp and −1080 to −781 bp; and to the promoter of *Serpinb2* at −2517 to −2221 bp and −1320 to −1021 bp (Supplementary Fig. 5a, b). We chose *pCcl4-2* and *pSerpinb2-8*, the promoter regions with the highest binding signals, for further validation. Dual-luciferase reporter assay

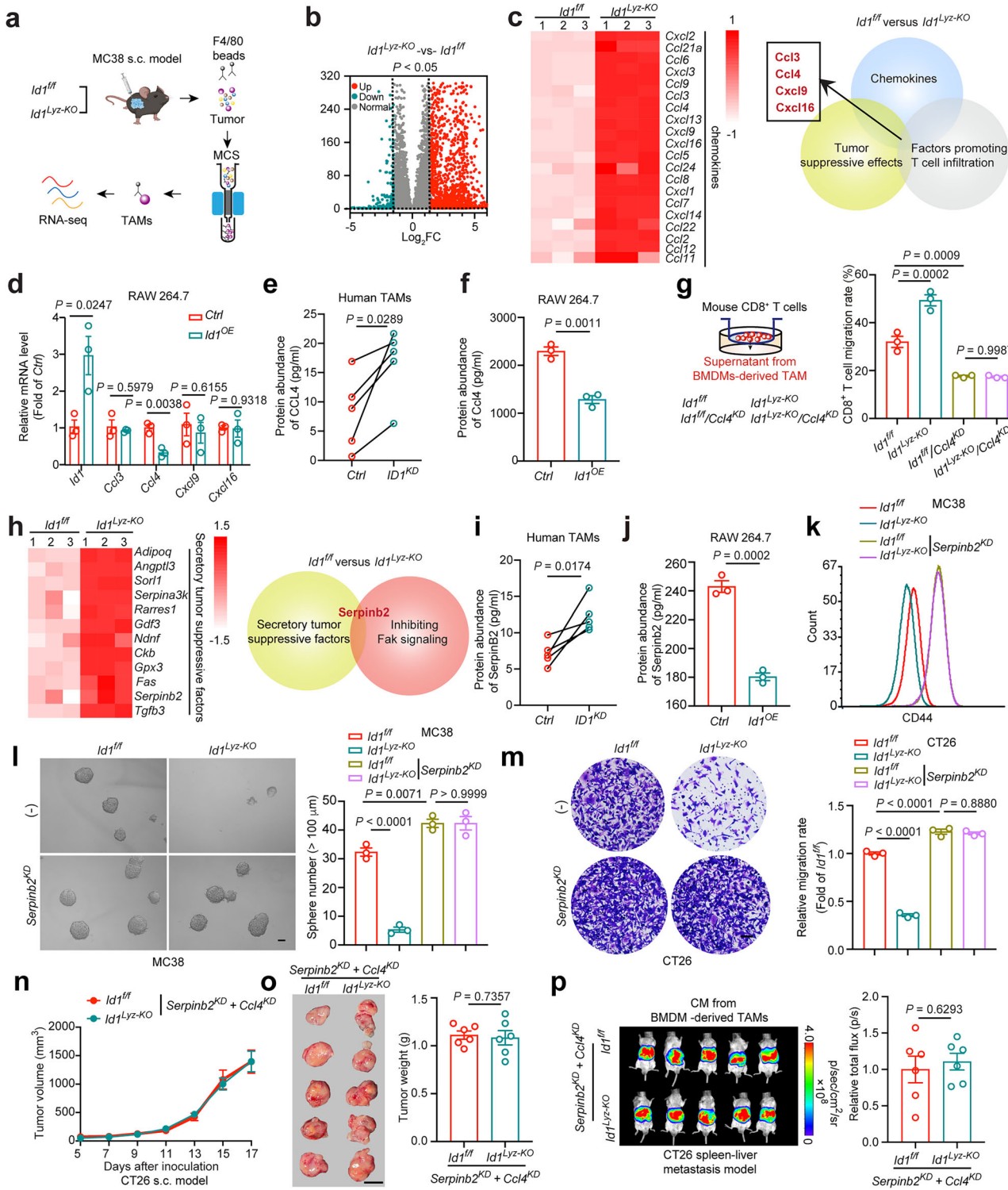

demonstrated that Id1 inhibited the transcription of *Ccl4* and *Serpinb2* through binding at *pCcl4-2* and *pSerpinb2-8* promoter regions (Fig. 6f).

To further explore the molecular mechanism by which ID1 regulates STAT1 transcriptional activity, we deeply dissected the binding domains of the two proteins. Co-IP assays showed that the N-terminus and the HLH domain of ID1 mediated its interaction with STAT1 (Supplementary Fig. 5c). In addition, the N-terminus, the DNA binding domain, and the SH2 domain of STAT1 mediated its association with ID1 (Supplementary Fig. 5d). The N-terminus and the SH2 domain have been reported to mediate STAT1 dimerization[39]. However, ID1 overexpression showed no effect on STAT1 homodimerization

(Supplementary Fig. 5e). The cytoplasmic and nuclear localization of STAT1 is a dynamic process to maintain moderate signaling activation. We thus questioned whether ID1 is involved in the regulation of STAT1 nuclear exportation. Overexpression of *Id1* reduced the basal Stat1 and p-Stat1 abundance in the nucleus under the treatment of IFN-γ but increased them in the cytoplasm (Fig. 6g). CRM1 recognizes a region in the DNA binding domain of STAT1 to redistribute it back to the cytoplasm[39]. We found that ID1 enhanced the interaction of CRM1 with STAT1 (Fig. 6h, i). Co-IP assays further confirmed that all three proteins interact with each other, implying a heterotrimeric protein complex formed among ID1, STAT1, and CRM1 (Supplementary Fig. 5f).

**Fig. 5 | ID1 in TAMs mediates tumor immune evasion and CRC stemness maintenance by inhibiting *CCL4* and *SerpinB2* transcription. a** Schematic diagram of the transcriptome sequencing. **b** Volcano plot of differentially expressed genes of the indicated groups presented in (**a**), $n = 3$ biologically independent samples per group in the RNA-seq data. **c** Log2FC heatmap of differentially expressed chemokines encoding genes (left). Schematic diagram for the screening of Ccl3, Ccl4, Cxcl9, and Cxcl16 (right). **d** Relative mRNA level of *Ccl3, Ccl4, Cxcl9* and *Cxcl16* in different groups of RAW264.7 cells, $n = 3$ biologically independent samples, Student's *t*-test. **e** CCL4 abundance in the CM from *Ctrl* and *ID1^KD* TAMs, $n = 5$ patients, paired *t*-test. **f** Ccl4 abundance in the CM from different groups of RAW 264.7 cells, $n = 3$ biologically independent samples, Student's *t*-test. **g** CD8+ T cells migration under the attraction of CM from *Id1^f/f* or *Id1^Lyz-KO* BMDM-derived TAMs with or without *Ccl4* knocking down, $n = 3$ biologically independent samples, one-way ANOVA test. **h** Log2FC heatmap of differentially expressed genes encoding secretory tumor suppressive factors (left). Schematic diagram for the screening of Serpinb2 (right). **i** SerpinB2 abundance in the CM from *Ctrl* and *ID1^KD* TAMs, $n = 5$ patients, paired *t*-test. **j** Serpinb2 abundance in the CM from different groups of RAW 264.7 cells, $n = 3$ biologically independent samples, Student's *t*-test. **k** Flow cytometry analysis of CD44 in MC38 cells pretreated with CM from *Id1^f/f* or *Id1^Lyz-KO* BMDM-derived TAMs, which were infected with or without *Serpinb2*-specific shRNA lentivirus (*Serpinb2^KD*), $n = 3$ biologically independent samples. **l, m** Images and quantification of MC38 tumor spheres (**l**), and invasiveness of indicated CT26 cells (**m**) in the indicated groups as mentioned in (**k**), $n = 3$ biologically independent samples, one-way ANOVA test. Scale bar, 100 µm. **n, o** Tumor volumes (**n**), representative tumor images, and tumor weight (**o**) of the indicated groups in CT26 s.c. model, $n = 6$ mice per group, Student's *t*-test. **p** Representative bioluminescence images and the statistical data of indicated groups in CT26 spleen-liver metastasis model, $n = 6$ mice per group, Student's *t*-test. Source data are provided as a Source Data file.

Leptomycin B is a CRM1 inhibitor which keeps the nuclear distribution of Stat1. Under IFN-γ treatment, ID1 redistributed the STAT1 dimer from the nucleus to the cytoplasm, which could be abrogated by leptomycin B treatment (Fig. 6j). Beisdes, we found that *Id1* depletion did not affect the binding ability of Stat1 with *Serpinb2* and *Ccl4* promoters under the treatment of leptomycin B. (Supplementary Fig. 5g). These data suggested that ID1 promotes the recruitment of CRM1 to STAT1, which facilitates STAT1 cytoplasmic distribution and inhibits STAT1-induced *CCL4* and *SerpinB2* transcription (Supplementary Fig. 5h).

## Targeting ID1 inhibits CRC progression and sensitizes tumor cells to chemotherapy and immunotherapy

ID1 is a short-living protein and is mainly subjected to ubiquitination-mediated proteasomal or autophagic degradation[40,41]. We explored whether pharmacological reduction of ID1 could eliminate the tumor-promoting role of TAMs. Ubiquitination-specific protease 1 (USP1) is a deubiquitinase that has been reported to keep ID proteins stable[42]. We evaluated the potential antitumor effect of ML323, a selective inhibitor of USP1, in both a syngeneic s.c. CRC mouse model and a syngeneic orthotopic metastasis CRC mouse model. In the s.c. model, ML323 administration reduced ID1 expression in TAMs and manifested a marked tumor inhibition effect, as shown by the tumor volume and tumor weight compared with the vehicle group (Fig. 7a, b; Supplementary Fig. 6a–c). In addition, we detected increased tumor-infiltrating CD8+ T cells and enhanced CD8+ T-cell effector function as assessed by Granzyme B and IFN-γ expression in tumor-infiltrating CD8+ T cells from the ML323-treated group (Fig. 7c; Supplementary Fig. 6d, e). In addition, tumor cells isolated from tumor tissues of the ML323-treated group showed reduced expression of CSC markers (Fig. 7d). In the orthotopic liver metastasis model, ML323 treatment obviously alleviated liver metastasis, in which CD8+ T cell infiltration and effectiveness were enhanced, and tumor cell stemness traits were inhibited (Fig. 7e–i; Supplementary Fig. 6f, g).

We further confirmed that ML323 inhibited tumor progression partly through regulating ID1 expression in TAMs. Using *Id1^f/f* and *Id1^Lyz-KO* mice, we demonstrated that depletion of *Id1* in myeloid cells largely but not totally reversed the tumor inhibitory role of ML323 in MC38 s.c. model (Fig. 7j, k; Supplementary Fig. 6h), indicating that ML323 works partly through Id1 in myeloid cells. TAMs were then isolated from tumor tissues in a vehicle or ML323-treated mice to study the influence of ML323 (Fig. 7l). Using a non-contact coculture system, we demonstrated that TAMs treated with ML323 promoted CD8+ T cells migration and inhibited CD44 and Aldh expression in CT26 cells (Fig. 7m, n). In addition, in vitro assays further confirmed the direct influence on TAMs by ML323. We validated that CM from BMDM-derived TAMs with ML323 treatment also promoted CD8+ T cell migration ability, decreased CSC marker expression, tumor sphere-forming ability, and invasiveness. Moreover, *Id1^Lyz-KO* in TAMs abrogated the effects of ML323 treatment on cancer stemness and CD8+ T cell migration (Fig. 7o, p; Supplementary Fig. 6i, j), indicating the effect of ML323 on TAMs are mediated by Id1. Mechanistically, protein abundance of Serpinb2 and Ccl4 were both upregulated in the ML323 treated TAMs (as presented in Fig. 7l) (Supplementary Fig. 6k, l). We conducted in vitro experiments using RAW 264.7 cells to further validate that ML323's impact on Serpinb2 and Ccl4 is through the regulation of Id1. Our results demonstrated that ML323 treatment led to an increase in the expression of Serpinb2 and Ccl4 in RAW 264.7 cells which was negated by *Id1* overexpression (Fig. 7q, r). These data together indicated that ML323 exerts antitumor efficacy by reducing the expression of Id1 in TAMs.

These results prompted us to evaluate the potential effect of ML323 (at a lower dose of 3 mg/kg) in combination with chemotherapy or immunotherapy in preclinical murine models. First, we treated CT26 tumor-bearing mice with ML323 in combination with 5-fluorouracil (5-FU) (Fig. 7s). ML323 in combination with 5-FU manifested a synergistic antitumor effect, as shown by the tumor volume (Fig. 7t) and tumor weight (Fig. 7u, v) compared with the 5-FU or ML323 single agent treatment groups. Second, we determined the synergy of ML323 with anti-CTLA4 in CT26 tumor-bearing mice (Fig. 7w). The combination treatment resulted in a stronger antitumor effect than ML323 or anti-CTLA4 alone (Fig. 7x–z). In addition, increased CD8+ T cell infiltration and enhanced CD8+ T cell effector function were observed in the combined group (Supplementary Fig. 6m, n). These data raised the possibility that targeting ID1 through inhibiting USP1 activity may be a potential strategy for improving the efficacy of chemotherapy and immunotherapy for CRC treatment.

As all the experiments have been performed in male mice, we further studied whether there is any sexual difference in the role of ID1 in TAMs. MC38 tumor-bearing female mice were intratumorally injected with CM derived from female *Id1^f/f* or *Id1^Lyz-KO* TAMs (Supplementary Fig. 7a). Tumors injected with CM from *Id1^Lyz-KO* TAMs grew slower than those injected with CM from *Id1^f/f* TAMs as observed in the male mice (Supplementary Fig. 7b–d). Besides, more CD8+ T cell infiltration, higher CD8+ T cell activity, and lower CD44^high Lgr5+ expression in the CD45- Epcam+ tumor cells could be observed in the tumor tissues of the *Id1^Lyz-KO* group (Supplementary Fig. 7e–h). These data indicated that the tumor-promoting role of ID1 in TAMs has no difference between sexes. To explore the generalizability of our findings, two other s.c. tumor mouse models using H22 (mouse hepatocellular carcinoma cell line) and Pan02 (mouse pancreatic ductal adenocarcinoma cell line) were established. Depletion of *Id1* in BMDM-derived TAMs also inhibited tumor growth of liver cancer and pancreatic cancer, decreasing the tumor stem cell proportion and promoting the infiltration and activity of CD8+ T cells (Supplementary Fig. 7i–v). These data suggested that our findings on ID1 expressing TAMs in the promotion of tumor progression may have broad relevance in other cancer types.

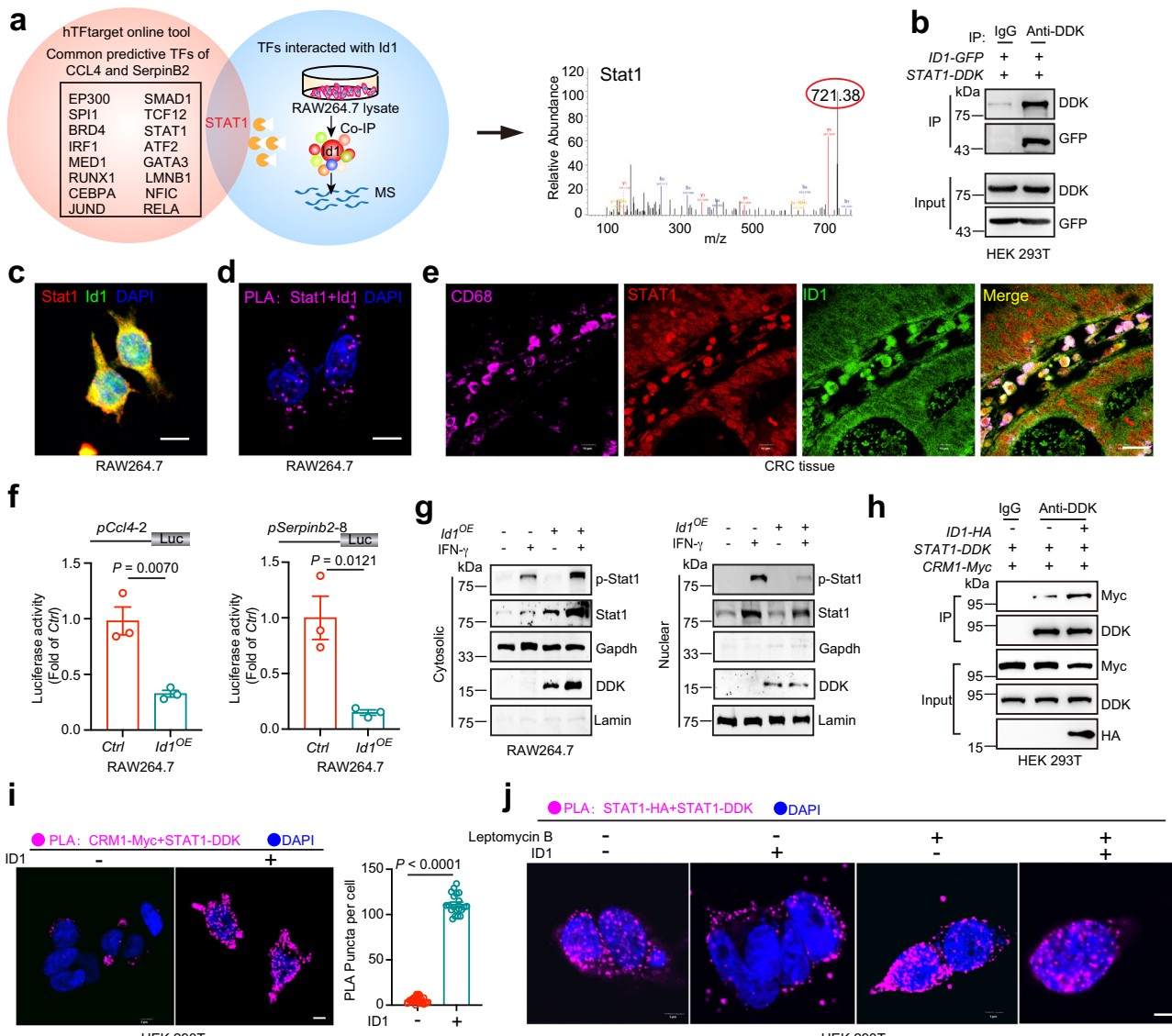

**Fig. 6 | ID1 interacts with STAT1 to inhibit the transcription of *CCL4* and *SerpinB2*. a** Screening for ID1-interacting transcription factors responsible for *CCL4* and *SerpinB2* transcription. **b** Co-immunoprecipitation (Co-IP) of ID1 with STAT1 in HEK 293T cells transfected with *STAT1-DDK* and *ID1-GFP* expressing plasmids. $n = 1$ biologically independent sample. **c** Immunofluorescent staining of Id1 and Stat1 in TAM-like RAW264.7 cells. Scale bar, 10 μm, $n = 3$ biologically independent samples. **d** Proximity ligation assay (PLA) for defining the interaction between Id1 and Stat1 in TAM-like RAW264.7 cells. Scale bar, 10 μm, $n = 10$ biologically independent samples. **e** Immunofluorescence staining of ID1, STAT1 and CD68 in tumor tissue from CRC patients. Scale bar, 20 μm, $n = 3$ biologically independent samples. **f** Analysis of the luciferase activity of truncated promoter sequences of *Ccl4* and *Serpinb2* in *Ctrl*

or *Id1^{OE}* RAW264.7 cells, $n = 3$ biologically independent samples, Student's *t*-test. **g** Effects of Id1 on Stat1 and p-Stat1 subcellular localization under IFN-γ stimulation in RAW264.7 cells, $n = 3$ biologically independent samples. **h** The effects of ID1 on the interaction between STAT1 and CRM1, $n = 3$ biologically independent samples. **i** Representative images of PLA and the statistical data showing the protein interaction between STAT1 and CRM1 with ID1 ectopic expression or not, $n = 20$ biologically independent cells. Scale bar, 5 μm. **j** Representative images of PLA showing the effects of ID1 on the dimerization and subcellular localization of STAT1 under the treatment of leptomycin B (100 nM), Scale bar, 5 μm, $n = 3$ biologically independent samples. Source data are provided as a Source Data file.

## Discussion

Macrophages are highly plastic cells, which undergo diverse forms of functional activation in response to different stimuli. Stress and inflammatory signals in the TME may remodel the function of infiltrating macrophages to mediate tumor immune evasion and sustain cancer stemness. A network of transcription factors underlies the plasticity and polarization of macrophages[5]. Our study revealed that the TME endowed TAMs with high ID1 expression, which interacts with STAT1 to inhibit *CCL4* and *SerpinB2* transcription, two STAT1 target genes, leading to CD8^+ T cell exclusion in tumor sites and cancer stemness traits maintenance (Supplementary Fig. 8). Such effects, in turn, promote CRC growth and metastasis. TAMs with high ID1

expression predict poor outcomes and low susceptibility to chemotherapy or immunotherapy for CRC patients. Besides CRC, our findings also have broad relevance in liver cancer and pancreatic cancer. Our study revealed the biological and clinical significance of ID1 in TAMs in the induction and maintenance of alternative macrophage activation, highlighting the therapeutic potential for targeting ID1 in the phenotypic switch of TAMs from immunosuppressive to proinflammatory, activation of T cell responses and the delay of tumor growth and metastasis.

Although ID1 is known to play pivotal roles in myeloid development[43], the ID1 connection in remodeling the tumor immune microenvironment is scarcely understood. Previous

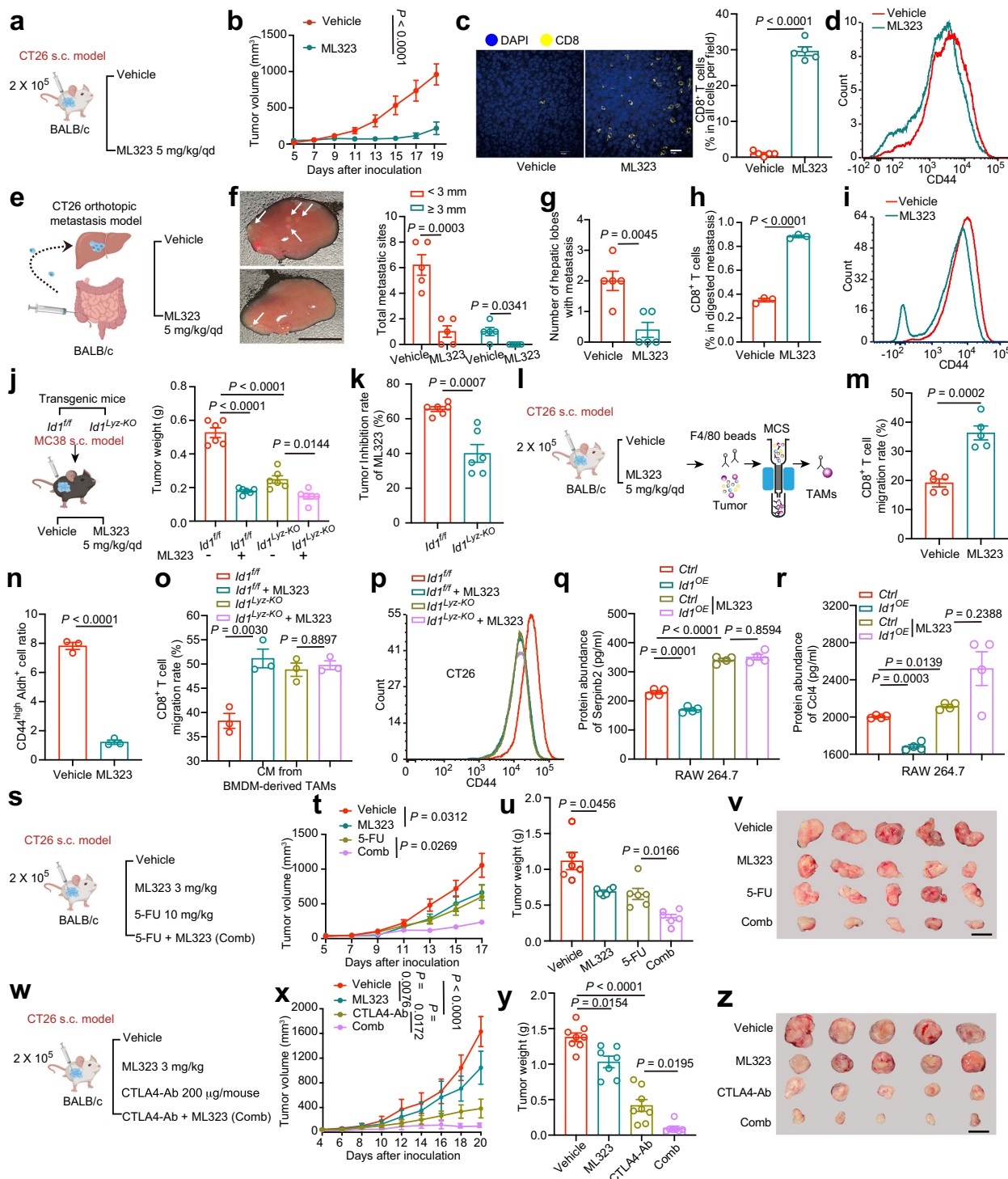

studies found that ID1 is upregulated by some tumor-derived factors, such as transforming growth factor beta (TGFβ), in bone marrow-derived myeloid cells to promote the myeloid cell differentiation switch from dendritic cells (DCs) to MDSCs via downregulating *Irf8* during tumor progression[22]. Such effect results in the formation of tumor immunosuppressive microenvironment including expansion of T-regs and hampered proliferation and activation of CD8[+] T cells[22]. Here, our study demonstrated that ID1 is highly expressed in CRC TAMs, which drives the loss of the antitumorigenic phenotype by changing the expression of two secretory proteins of TAMs. From the cancer cell side, TAMs are known to protect CSCs from hostile environment[44]. Here, TAMs with high ID1

expression sustain CSC stemness traits by creating a protective niche wherein these cells comfortably reside and acquire metastatic capacity. Such effects arise from the reduced secretion of SerpinB2, a critical indicator of stem cell toxicity[45]. Loss of SerpinB2 enhances CSC survival and expansion by activating FAK-YAP signaling. From the effector T cell side, TAMs inhibit the antitumor immune response by impeding CD8[+] T cells from reaching tumor sites or impairing their cytotoxic activity[46,47]. TAMs with high ID1 expression exclude CD8[+] T cell infiltration by inhibiting the secretion of CCL4 from TAMs. Interestingly, depletion of *Id1* in MDSCs had no influence on CD8[+] T cell migration, suggesting that ID1 plays distinct roles in the proliferation/activation or migration ability of CD8[+]

**Fig. 7 | Targeting ID1 inhibits CRC progression and sensitizes tumor cells to chemotherapy and immunotherapy. a** Schematic diagram for evaluating the therapeutic effects of ML323. **b, c** Tumor volumes (**b**), representative images (left) and quantification (right) of CD8$^+$ T cell infiltration (**c**) of the indicated groups presented in (**a**), $n = 8$ (**b**), or 5 (**c**) mice per group, Welch's test. Scale bar, 20 µm. **d** CD44 expression in the indicated groups. **e** The therapeutic effects of ML323 in the liver metastasis model. **f–i** Representative gross images of metastatic sites (**f**), number of hepatic lobes with metastatic tumor (**g**), the percentage of CD8$^+$ T cells (**h**), and flow cytometry of CD44 in CD45$^-$ Epcam$^+$ cells (**i**) of the indicated groups presented in (**e**). $n = 5$ mice per group in **f–h**, Welch's test. $n = 3$ biologically independent samples in (**i**), Student's *t*-test. Scale bar, 1 cm. **j, k** Therapeutic effects of ML323 in *Id1$^{f/f}$* or *Id1$^{Lyz-KO}$* tumor-bearing mice. Tumor weight (**j**) and tumor inhibition rate (**k**) were shown, $n = 6$ mice per group, one-way ANOVA test (**j**), Student's *t*-test (**k**). **l** Schematic diagram for isolating TAMs from indicated mice. **m** The migration rate of CD8$^+$ T cells cocultured with TAMs isolated as presented in (**l**), $n = 5$ mice per group. **n** CD44$^{high}$ Aldh$^+$ cell ratio of CT26 cells cocultured with TAMs

isolated as presented in (**l**), $n = 3$ mice per group. **o** Migration of CD8$^+$ T cells when cocultured with *Id1$^{f/f}$* or *Id1$^{Lyz-KO}$* TAMs under ML323 (10 µM) treatment, $n = 3$ biologically independent samples, one-way ANOVA test. **p** CD44 expression in CT26 cells cocultured with *Id1$^{f/f}$* or *Id1$^{Lyz-KO}$* TAMs under ML323 (10 µM) treatment, $n = 3$ biologically independent samples. **q, r** Effects of ML323 on expression of Serpinb2 (**q**) and Ccl4 (**r**) in the indicated groups, $n = 4$ biologically independent samples. **s** Therapeutic effects of ML323 with 5-FU. Comb: ML323 + 5-FU. **t–v** Tumor volumes (**t**), tumor weight (**u**), and representative tumor images (**v**) of indicated groups in (**s**), $n = 6$ mice per group, one-way ANOVA test. Scale bar, 1 cm. **w** Therapeutic effects of ML323 with CTLA4-Ab. Comb: ML323 + CTLA4-Ab. **x** Tumor volumes of indicated groups in (**w**), $n = 6$ mice per group, one-way ANOVA test. **y** Tumor weight of indicated groups in (**w**), $n = 8$, 7, 8 and 7 in the groups of Vehicle, ML323, CTLA4-Ab, and Comb, one-way ANOVA test. **z** Representative tumor images of indicated groups in (**w**). Scale bar, 1 cm. Elements of **a, e, j, l, s,** and **w** are created with BioRender.com. Source data are provided as a Source Data file.

T cells between MDSCs and TAMs. Our discovery suggests that TAMs with high ID1 expression establish a malevolent alliance between the immunosuppressive TME and CSCs to facilitate tumor progression.

Previous studies have established that inhibition of DNA binding is the canonical function of ID proteins based on their extensive sequence homology in the HLH motif, which mediates the formation of nonfunctional heterodimers with bHLH transcription factors, impairing DNA binding and bHLH-directed transcription[32]. ID proteins also exert inhibitory activity toward non-bHLH transcription factors, such as ETS and paired box (PAX) families. IDs bind to the ETS DNA-binding domain or the paired DNA-binding domain via the HLH motif and disrupt protein–DNA interactions[48,49]. In this study, we proposed an action paradigm for the suppressive role of ID1 with respect to STAT1 transcriptional activity. ID1 shows no effect on the binding ability of STAT1 with the promoter of *CCL4* and *SerpinB2* but acts as an adaptor protein to support the accessibility of STAT1 to the CRM1 export carrier, which promotes STAT1 cytoplasmic distribution and inhibits STAT1-mediated target gene transcription. In fact, ID1 was reported to interact with the non-transcription factor protein Caveolin-1 (Cav-1) to induce AKT activation by acting as an adaptor protein[50]. The physical interaction between ID1 and Cav-1 promotes the binding of Cav-1 with phosphatase 2A (PP2A), a dephosphorylating enzyme of AKT, to suppress PP2A enzyme activity. Here, we provide evidence to indicate that ID1 in TAMs can also play a scaffolding role to impede CD8$^+$ T cell recruitment and activate the FAK-YAP cascade in cancer cells by impairing STAT1-mediated transcription of *CCL4* and *SerpinB2*. Our findings have introduced an additional layer of understanding to the divergent functions of ID1 in cancer biology by highlighting the crosstalk between TAMs and cancer cells.

From the therapeutic angle, both cancer stem cells and pro-tumorigenic TAMs are addicted to ID1, suggesting that ID1 may be a promising anticancer target. Here, we found that ML323 administration eliminates cancer stem cells increases CD8$^+$ T cells infiltration simultaneously, and enhances the therapeutic efficacy of anti-CTLA4 antibody or 5-FU, demonstrating the great potential of ID1 as a dual-functional target. Notably, ID1 expression is usually low in adults under homeostatic conditions[51]. Such expression features suggest that targeting ID1 may result in fewer side effects in healthy tissues.

Overall, our work identifies TAM-expressed ID1 as a central molecular node, dually controlling the cancer initiation capacity and inducing tumor immune evasion. High ID1 expression in TAMs represents an immune evasion and cancer stemness-maintaining mechanism in CRC. Additionally, we provide proof of the principle that targeting ID1 stability may enhance immunotherapy and chemotherapy sensitivity in patients with CRC.

## Methods

### Study approval

All animal procedures were conducted in accordance with the guidelines of the Institutional Committee in the Institute of Materia Medica, Chinese Academy of Medical Sciences & Peking Union Medical College, and Chinese Center for Disease Control and Prevention for the Ethics of Animal Care and Treatment. The animal study also accorded with the ARRIVE guidelines. According to the requirements of the ethics committee, the maximal tumor volume allowed was 2000 mm$^3$. In some cases, this limit has been exceeded by the last day of measurement and the mice were immediately euthanized. The CRC tumor samples were collected with approval by the institutional review board of Cancer Hospital at the Chinese Academy of Medical Sciences or by the Shanghai Outdo Biotech Ethics Committee. Informed consent was obtained from the patients.

### Human tumor samples

Human CRC tissue microarray (HcolA180Su17) was purchased from Shanghai Outdo Biotech. Fresh human CRC tumor tissues were provided by the Cancer Hospital Chinese Academy of Medical Sciences (Supplementary Table 2). The tumor-promoting role of ID1 in TAMs has no difference between males and females when taking sex into account in our study. Human CRC tissue microarray analysis indicated that the expression of ID1 in TAMs has no correlation with sex.

### Animal studies

C57BL/6J mice (male and female, 6-8 weeks old), BALB/c mice (male, 6–8 weeks old), BALB/c-*Foxn1$^{nu}$*/Nju (named as BALB/c nude hereafter) mice (male, 6 weeks old) were purchased from the BEIJING HFK BIOSCIENCE CO., LTD. OT1 and *Lyz2$^{tm1(cre)Ifo/J}$* mice were provided by Cyagen Biosciences Inc. *Id1$^{f/f}$* mice were generated by Cyagen Biosciences Inc. Myeloid cell-lineage-specific *Id1* deficient mice (*Id1$^{Lyz-ko}$*) were generated by crossing *Id1$^{f/f}$* mice with *Lyz2$^{tm1(cre)Ifo/J}$* mice. All mice were maintained in the animal facility at the Institute of Materia Medica and Chinese Center for Disease Control and Prevention under specific-pathogen-free (SPF) conditions. For animal studies, the mice were earmarked before grouping and randomly separated into groups by an independent person. We used 6 to 10 mice per experimental group in all animal experiments. The tumor-promoting role of ID1 in TAMs has no difference between males and females when taking sex into account in our study.

For the subcutaneous tumor model, 100 µL matrigel (354230, Corning)/tumor cell suspension containing $1 \times 10^6$ MC38 cells, $2 \times 10^5$ CT26 cells, $5 \times 10^5$ H22 cells, or $1 \times 10^6$ Pan02 cells were subcutaneously inoculated into the right flank of each mouse to establish the syngeneic tumor mouse models. Tumor growth was monitored twice weekly with a caliper. Tumor volume ($T_V$) was calculated using the formula: $T_V = 0.5 \times L \times W^2$.

For the CT26 orthotopic tumor model, 50 μL matrigel/tumor cell suspension containing $1 \times 10^6$ CT26-luc cells was injected into the cecal wall. In some experiments, 100 μL conditional medium (CM) from different groups of TAMs was administered by intraperitoneal injection every other day. About 25 days after tumor cell inoculation, tumor conditions were inspected through a bioluminescence imaging system.

For the orthotopic liver metastasis tumor model, $1 \times 10^6$ CT26 cells were injected into the cecal wall of BALB/c mice, about 1/10 mice will develop liver metastasis 30 days after inoculation. The metastatic tumor tissues were then isolated and digested into single-cell suspension, which was injected into the cecal wall of BALB/c mice in the second round. In this round, 100% of mice will develop liver metastasis. 100 μL CM from different groups of TAMs was administered by intraperitoneal injection every other day. About 20 days after tumor cell inoculation, tumor conditions were inspected.

For the CRC spleen-liver metastasis model, 50 μL tumor cell suspension containing $8 \times 10^5$ CT26-luc cells was prepared in phosphate-buffered saline and injected into the spleen of mice. The blood vessels on the injection side need to be ligated to prevent orthotopic tumor growth in the spleen. Tumor-bearing mice were treated with CM from $Id1^{f/f}$ and $Id1^{Lyz-KO}$ TAMs every other day starting on the third day after tumor cell inoculation. About 20 days after tumor cells inoculation, mice were inspected through a bioluminescence imaging system.

For the CRC lung metastasis model, 50 μL tumor cell suspension containing $5 \times 10^5$ CT26-luc cells was prepared in phosphate-buffered saline and injected into the tail vein of mice. Tumor-bearing mice were treated with CM from $Id1^{f/f}$ and $Id1^{Lyz-KO}$ TAMs every other day starting on the third day after tumor cell inoculation. About 15 days after tumor cells inoculation, mice were inspected through a bioluminescence imaging system.

For the CD8+ T cells deletion assay, $5 \times 10^5$ CT26 cells were subcutaneously inoculated into BALB/c mice. CM from $Id1^{f/f}$ and $Id1^{Lyz-KO}$ TAMs were given intratumorally every other day. Meanwhile, anti-CD8β antibodies were administered every 3 days from the day before the s.c. inoculation until the endpoint. IgG isotype antibody was used as a control.

For the Y15 treatment assay, CT26-bearing BALB/c nude mice were divided into 4 groups and treated with CM from Ctrl or $Id1^{OE}$ M1-like RAW264.7 cells intratumorally every other day. Meanwhile, two groups treated with CM from Ctrl or $Id1^{OE}$ M1-like RAW264.7 cells were further administered with Y15 (5 mg/kg/day) until the endpoint.

For the therapeutic study in the CT26 s.c. model, CT26 cells were subcutaneously inoculated into the right flank of BALB/c mice. Three days after cancer cells inoculation, mice were administered ML323 (S7529, Selleck, 5 mg/kg, i.p.) once a day until the endpoint. For the combinational therapy with 5-FU, mice were treated with ML323 (3 mg/kg, i.p.) once a day; or 10 mg/kg 5-FU (S1209, Selleck, i.p.) on the day of 8, 11, 14 after cancer cells inoculation; or in the combination of the two agents until the endpoint. For the combinational therapy with anti-CTLA-4 antibody (BE0164, Bio X Cell), mice were treated with ML323 (3 mg/kg, i.p.) once a day; or CTLA-4-Ab (200 μg/mice, i.p.) on the day of 6, 9, 12 after cancer cells inoculation; or in the combination of the two agents until the endpoint.

For the therapeutic study in the orthotopic metastasis model, $1 \times 10^6$ CT26 cells were injected into the cecal wall of BALB/c mice, about 1/10 mice will develop liver metastasis 30 days after inoculation. The metastatic tumor tissues were then isolated and digested into single-cell suspension, which was injected into the cecal wall of BALB/c mice in the second round. In this round, 100% of mice will develop liver metastasis. Half of the tumor-bearing mice will be treated with ML323 (5 mg/kg) 3 days after surgery. About 20 days after tumor cells inoculation, tumor conditions were inspected.

The in vivo macrophage adoptive transfer was performed as previously reported[25]. Briefly, TAMs were isolated from MC38-derived tumors inoculated in donor mice by magnetic beads-based single-cell isolation. Additionally, for some experiments, primary BMDMs were polarized into TAMs under the stimulation of IL-4 (214-14, Peprotech, 20 ng/mL) for 48 h and harvested as single-cell suspension. Purified TAMs were then mixed with MC38 cells in a ratio of 1:3, and $7.5 \times 10^5$ total cells were injected subcutaneously into new recipient mice.

## Antibodies
c-MYC-monoclonal antibody (Proteintech, cat. #67447-1-Ig, clone 3D9C12), Anti-TCF7L2 antibody (Abcam, cat. #ab32873, clone 0.T.149), Anti-Id2 antibody (Santa Cruz, cat. #sc-398104, clone E-7), Snail (C15D3) rabbit mAb antibody (Cell Signaling Technology, cat. #3879S, clone N/A), ID1 polyclonal antibody (Proteintech, cat. #18475-1-AP), Anti-CD68 antibody (Abcam, cat. #ab955, clone KP1), Anti-F4/80 antibody (Abcam, cat. #ab6640), Mouse-anti-GAPDH mAb (ZSGB-BIO, cat. #TA-08, clone OTI2D9), FAK Antibody (Cell Signaling Technology, cat. #3285S), Phospho-FAK (Tyr397) antibody (Cell Signaling Technology, cat. #3283S), STAT1 polyclonal antibody (Proteintech, cat. #10144-2-AP), Phospho-Stat1 (Tyr701) (58D6) rabbit mAb (Cell Signaling Technology, cat. #9167S, clone 58D6), Lamin A/C (4C11) mouse mAb (Cell Signaling Technology, cat. #4777S, clone 4C11), Anti-GFP (rabbit) (MBL Biotech, cat. #598), HA tag polyclonal antibody (Proteintech, cat. #51064-2-AP), Anti–Myc-tag pAb (MBL Biotech, cat. #562), Anti-DDDDK-tag mAb (MBL Biotech, cat. #M185-7), DYKDDDDK tag monoclonal antibody (Proteintech, cat. # 66008-3-Ig, clone 2B3C4), Anti-CD8α (mouse specific) (Cell Signaling Technology, cat. #98941S, clone D4W2Z), STAT1 monoclonal antibody (Invitrogen, cat. #AHO0832, clone STAT1-79), Anti-STAT1 antibody [EPR23049-111]–ChIP grade (Abcam, cat. #ab239360), Anti-Ki67 antibody (Abcam, cat. #ab15580), APC anti-mouse/human CD44 antibody (Biolegend, cat. #103011, clone IM7), PE anti-mouse CD45 recombinant antibody (Biolegend, cat. #157604, clone QA17A26), APC anti-mouse CD45 antibody (Biolegend, cat. #103111, clone 30-F11), FITC anti-mouse CD45 antibody (Biolegend, cat. #157607, clone QA17A26), PE anti-mouse CD326 (Ep-CAM) Antibody (Biolegend, cat. #118205, clone G8.8), PerCP/Cyanine5.5 anti-mouse CD3ε antibody (Biolegend, cat. #100328, clone 145-2C11), APC/Cyanine7 anti-mouse CD8α antibody (Biolegend, cat. #100714, clone 53-6.7), PE anti-mouse IFN-γ antibody (Biolegend, cat. #505808, clone XMG1.2), PE/Cy7 anti-human/mouse Granzyme B recombinant antibody (Biolegend, cat. #372213, clone QA16A02), PE anti-mouse F4/80 recombinant antibody (Biolegend, cat. #157304, clone QA17A29), Anti-mouse CD3 SAFIRE purified (BioGems, cat. #05112-25, clone 17A2), Anti-Mouse CD28 SAFIRE purified (BioGems, cat. #10312-25, clone 37.51), InVivoMAb rat IgG1 isotype control (Bio X Cell, cat. #BE0088, clone HRPN), InVivoMAb anti-mouse CD8β (Bio X Cell, cat. #BE0223, clone 53-5.8), InVivoMAb anti-mouse CTLA-4 (Bio X Cell, cat. #BE0164, clone 9D9), YAP (D8H1X) XP Rabbit mAb (Cell Signaling Technology, cat. #14074, clone N/A), Anti-YAP (phospho Y357) antibody (Abcam, ab254343, clone N/A), FITC anti-mouse Lgr5 antibody (Miltenyi Biotec, cat. #130-111-393, clone DA04-10E8.9), PE anti-human LGR5 antibody (BioLegend, cat. #373803, clone SA222C5), PE-Cyanine7 anti-human CD8α antibody (BioLegend, cat. #301012, clone RPA-T8), PE anti-mouse Ly-6G/Ly-6C (Gr-1) antibody (BioLegend, cat. #108407, clone RB6-8C5). For immunoblotting, antibodies were diluted as 1:1000. For immunostaining and immunohistochemistry, antibodies were diluted as 1:100. For flow cytometry, antibodies were added 1 μL per $10^6$ cells.

## Chemicals and recombinant proteins
The following chemicals and recombinant proteins were used: Y15 (Selleck, S5321), 5-Fluorouracil (5-FU) (Selleck, S1209), ML323 (Selleck, S7529), Propidium iodide (Solarbio, P8080), Matrigel (Corning, 354230), Fibronectin (Corning, 354008), StemXVivo serum-free tumorsphere media (R&D, CCM012), Ionomycin (Merck, IO634), GolgiStop (BD Biosciences, 554724), Brefeldin A solution (1,000X)

(Biolegend, 4200601), β-Mercaptoethanol (Solarbio, M8210), Human truStain FcX™ (Fc receptor blocking solution) (Biolegend, 422302), Recombinant murine IL-4 (Peprotech, 214-14), Recombinant murine IL-2 (Peprotech, 212-12), Recombinant human IFN-γ (Peprotech, 300-02), Recombinant murine IFN-γ (Peprotech, 315-05), Recombinant murine m-CSF (Peprotech, 315-02), Leptomycin B (Selleck, S7580), Phorbol 12-myristate 13-acetate (PMA) (Merck, P8139), Collagenase IV (Merck, C4-BIOC), Dnase I (Merck, 10104159001), Lithium phosphorus sulfide (LPS) powder (Merck, 916374), Puromycin (Invitrogen, A1113803), Fluorescent mounting medium with DAPI (ZSGB-BIO, ZLI-9557), Lymphocyte separation medium (Solarbio, P8900), 2,2,2-tribromoethanol (Merck, T48402), Protein A/G Plus-agarose (Santa Cruz, sc-2003), Ovalbumin (257-264) chicken (Merck, S7951), Verteporfin (Selleck, S1786).

## Commercial kits

The following commercial kits were used: CFSE cell division tracker kit (Biolegend, 423801), Precision count beads™ (Biolegend, 424902), SimpleChIP® plus sonication chromatin IP kit (Cell Signaling Technology, 56383), Four color labeled kit (TSA-RM) (PANOVUE Biotechnology Co.,LTD, 1100020), CD8⁺ T cell isolation kit, human (Miltenyi Biotec, 130-096-495), CD8⁺ T cells isolation kit, mouse (Miltenyl Biotec, 130-104-075), RNA-quick purification kit (Shang Hai Yishan Biotechnology Co., LTD, ES-RN001), Anti-PE microbeads (Miltenyi Biotec, 130-048-801), CD326 (EpCAM) microbeads, mouse (Miltenyi Biotec, 130-105-958), Cytofix/Cytoperm™ fixation/permeabilization solution kit (BD Biosciences, 554714), TransScrip@ One-Step RT-PCR superMix (Transgene, AT411-02), Secrete-pair gaussia luciferase assay kit (GeneCopoeia, LF062), Duolink® in situ detection reagents red (Merck, DUO92008), Mouse C−C motif chemokine 4 (Ccl4) ELISA kit (4A Biotech, MOEB0088), Human C−C motif chemokine 4 (CCL4) ELISA kit (4A Biotech, HUEB0132), Mouse plasminogen activator inhibitor 2, macrophage (Serpinb2) ELISA kit (4A Biotech, MOEB1945), Human plasminogen activator inhibitor 2 (SerpinB2) ELISA Kit (4A Biotech, HUEB2195), MycoBlue Mycoplasma Detector (Vazyme, D101-02), Dual luciferase reporter assay system (Promega, E1910), ALDEFLUORTM Kit (STEMCELL Technologies, 01700).

## Oligonucleotides

Quantitative polymerase chain reaction (qPCR) primers were all synthesized by Ruibotech company, and the sequences are as follows: Mouse *Ccl3* primers, Forward: ACTGCCTGCTGCTTCTCCTACA, Reverse: ATGACACCTGGCTGGGAGCAAA. Mouse *Cxcl9* primers, Forward: CCTAGTGATAAGGAATGCACGATG, Reverse: CTAGGCAGGTTT GATCTCCGTTC. Mouse *Cxcl16* primers, Forward: GCAGGGTACTTTG GATCACATCC, Reverse: AGTTCACGGACCCACTGGTCTT. Mouse *Ccl4* primers, Forward: CCCAGCTCTGTGCAAACCTA, Reverse: GAGCAA GGACGCTTCTCAGT. Mouse *Serpinb2* primers, Forward: ACCCAGA GAACTTCAGTGGCTG, Reverse: GAGAGAGGGAGAAGGCTGAATGG. Mouse *Gapdh* primers, Forward: ACCCAGAAGACTGTGGATGG, Reverse: CACATGGGGGGTAGGAACAC. Mouse *Id1* primers, Forward: T TGGTCTGTCGGAGCAAAGCGT, Reverse: CGTGAGTAGCAGCCGTTCA TGT. Human *CTGF* primers, Forward: CTTGCGAAGCTGACCTGGAAGA, Reverse: CCGTCGGTACATACTCCACAGA. Human *BIRC5* primers, Forward: CCACTGAGAACGAGCCAGACTT, Reverse: GTATTACAGGCGT AAGCCACCG. Human *DKK1* primers, Forward: GGTATTCCAGAAGA ACCACCTTG, Reverse: CTTGGACCAGAAGTGTCTAGCAC. Human *ITGB2* primers, Forward: AGTCACCTACGACTCCTTCTGC, Reverse: CA AACGACTGCTCCTGGATGCA. Mouse *Ctgf* primers, Forward: TGCG AAGCTGACCTGGAGGAAA, Reverse: CCGCAGAACTTAGCCCTGTATG. Mouse *Birc5* primers, Forward: CCTACCGAGAACGAGCCTGATT, Reverse: CCATCTGCTTCTTGACAGTGAGG. Mouse *Dkk1* primers, Forward: ATCTGTCTGGCTTGCCGAAAGC, Reverse: GAGGAAAATGGCT GTGGTCAGAG. Mouse *Itgb2* primers, Forward: CTTTCCGAGAGCA ACATCCAGC, Reverse: GTTGCTGGAGTCGTCAGACAGT.

## Viral particles

The following viral particles were all synthesized by HanBio Co., Ltd: Mouse *Ccl4* short hairpin RNA (shRNA) lentivirus, Mouse *Serpinb2* shRNA green fluorescent protein (GFP) lentivirus, Mouse *Id1* over-expression GFP lentivirus, Human *ID1* overexpression GFP lentivirus, Human *ID1* shRNA GFP lentivirus, Chicken *Ova* overexpression lentivirus.

## Cell lines

Human CRC cell lines HCT-116, DLD-1, and HCT-8; mouse CRC cell lines MC38 and CT26; mouse hepatocellular carcinoma cell line H22; mouse pancreatic ductal adenocarcinoma cell line Pan02; mouse monocytic/macrophage cell line RAW 264.7; human monocytic cell line THP-1; and human embryonic kidney 293T (HEK 293T) cells were purchased from the cell culture center of Peking Union Medical College, where cells were authenticated by STR profiling. DLD-1, MC38, CT26, H22, Pan02, RAW 264.7, and THP-1 cells were cultured in RPMI-1640 medium (GIBCO) supplemented with 10% fetal bovine serum (FBS). HEK 293T cells, HCT-116, and HCT-8 cells were cultured in DMEM medium (GIBCO) supplemented with 10% FBS. All cells were maintained at 37 °C in an incubator with a humidified atmosphere of 5% CO₂. Cells were regularly tested and verified to be mycoplasma negative using Myco-Blue Mycoplasma Detector (D101-02, Vazyme).

## Isolation of PMs, TAMs, and BMDMs

After i.p. injection of 5 mL DMEM cell culture medium containing 10% FBS, as well as penicillin and streptomycin, the peritoneal cells were collected in cell culture dishes. Two hours later, the floating cells were removed by washing the cells with PBS. The attached cells were considered as PMs (purity: ~90%) and were subjected to further experiments.

Mouse TAMs were isolated as follows. Tumors were isolated and minced in a Petri dish on ice and then enzymatically dissociated into single cells using Hanks balanced salt solution (HBSS) containing 200 U/ml collagenase IV (C4-BIOC, Merck) and 100 μg/mL Dnase I (10104159001, Merck) at 37 °C for 45 min. Cell suspension was filtered through a 70-μm cell strainer to exclude the cell aggregates. Red blood cells were lysed by re-suspending the pellet in 2 mL red blood cell lysis buffer and incubated at room temperature for 2 min. TAMs were sorted by F4/80 positive magnetic sorting (using PE-F4/80 antibody, 157304, Biolegend) and anti-PE-microbeads (130-048-801, Miltenyi Biotec)) according to the manufacturer's instructions. The isolated TAMs were cultured with RPMI-1640 medium supplied with 10% FBS.

Human TAMs were isolated as follows. Human CRC specimens were obtained from the Cancer Hospital Chinese Academy of Medical Sciences. Fresh tumor tissues through incubating in the digestion buffer (Miltenyi Biotec) for 30 min at 37 °C. Cell suspensions were filtered through a 70-μm cell strainer to exclude the cell aggregates. Red blood cells were lysed by re-suspending the pellet in 2 mL red blood cell lysis buffer and incubated at room temperature for 2 min. The cell suspensions were incubated overnight, and TAMs were adherent tightly. Cultures were shaken vigorously to detach non-adherent cells which were then removed from the culture by aspiration. Wash the remaining adherent cells three times with PBS by swirling the vessel and aspirating the supernatant. TAMs were isolated and cultured with Macrophage culture medium (12065074, Gibco) supplied with 10% fetal bovine serum, N2, B27, insulin, and m-CSF.

BMDMs were isolated as follows. Bone-marrow-derived cells were aseptically collected from 6 to 8-week-old mice by flushing the leg bones of euthanized mice with DMEM. Cell suspensions were filtered through a 70-μm cell strainer to exclude the cell aggregates. Red blood cells were lysed by re-suspending the pellet in 2 mL red blood cell lysis buffer and incubated at room temperature for 2 min. Approximately $1 \times 10^7$ bone-marrow-derived cells were purified by gradient centrifugation from the femurs and tibias of a single mouse. BMDMs were

cultured in DMEM with 10% FBS and treated with 20 ng/mL m-CSF (315-02, Peprotech) every other day.

## Immune profiling by flow cytometry

To quantify tumor-infiltrating lymphocytes and the expression of effector molecules in these cells, fresh tumor tissues were cut into small pieces and incubated in Collagenase IV (200 U/mL) and DNase I (100 μg/mL) in HBSS for 40 min at 37 °C and single-cell suspension was prepared. To analyze CD8+ T cell infiltration, anti-CD45 (103111, Biolegend), anti-CD3 (100328, Biolegend) and anti-CD8 antibodies (100714, Biolegend) were added in the single-cell suspension for 20 min. The cells were then washed and resuspended in magnetic-activated cell sorting (MACS) buffer (0.5% fetal calf serum, 2 mmol/L EDTA in phosphate-buffered saline). Percentage of CD8+ T cells CD8+ T cells (%) in total tumor = CD45+ T cells (%) × CD3+/CD45+ T cells (%) × CD8+/CD3+ T cells (%). To quantify effector T cell cytokine expression, T cells were enriched by density gradient centrifugation, and incubated in a culture medium containing PMA (P8139, Merck, 5 ng/ml), ionomycin (IO634, Merck, 500 ng/ml), brefeldin A (4200601, Biolegend, 1:1000) and GolgiStop (554724, BD Biosciences, 1:1000) at 37 °C for 4 h. Anti-CD45 (103111, Biolegend), anti-CD3 (100328, Biolegend), and anti-CD8 antibodies (100714, Biolegend) were added as described above for surface staining. The cells were then resuspended in 250 μL freshly prepared Fix/Perm solution (554714, BD biosciences) at 4 °C overnight. After being washed with Perm/Wash buffer, the cells were stained with anti-IFN-γ (505808, Biolegend) and anti-Granzyme B antibodies (372213, Biolegend) in MACS buffer for 30 minutes and washed with MACS buffer. All samples were run on a flow cytometer (BD FACS Verse) and analyzed by FCS Express 6 software. Gating strategies were summarized in Supplementary Fig. 9a–d.

## Plasmid construction

Full-length *ID1* and its truncated mutants, I1 (amino acids 1–52), I2 (amino acids 53–104), and I3 (amino acids 105–155) were constructed into pEGFP-C1 vector (CLONTECH Laboratories, Inc.). Full length of *STAT1* and its truncated mutants, S1 (amino acids 1–136), S2 (amino acids 137–317), S3 (amino acids 318–488), S4 (amino acids 489–576), S5 (amino acids 577–683) and S6 (amino acids 684–750) were constructed into pEGFP-C1 vector. *ID1-HA* and *STAT1-HA* were constructed into the pcDNA™3.1 vector. Promoter of *Ccl4* and *Serpinb2* were separately constructed into the pEZX-PG04 vector (GeneCopoeia, lnc.) respectively.

## Generation of stably expressing cell lines

To generate RAW264.7 cells stably expressing Id1 (or THP-1 cells stably expressing ID1), cells were infected with control or *Id1^OE* lentiviral particles. To generate MC38 cells stably expressing Ova, the cells were infected with *Ova*-expressing lentiviral particles. Briefly, the cells were mixed with the lentiviral particles and centrifuged at 1000×g, room temperature for 2 h. After 48 h of infection, cells stably expressing the relevant construct were selected in a medium containing 2 μg/mL puromycin (A1113803, Invitrogen) for 14 days or followed with GFP-positive cell sorting. After 2–3 passages selection in the presence of puromycin, the cultured cells were used without single-cell cloning.

## RNA extraction and real-time polymerase chain reaction

Total RNA was extracted using an RNA-Quick purification kit (ES-RN001, Shang Hai Yishan Biotechnology Co., Ltd) according to the manufacturer's instructions. RNA was quantified using a NanoDrop spectrophotometer. Reverse transcription of the total cellular RNA was carried out using oligo (dT) primers and MMLV reverse transcriptase. Polymerase chain reaction was performed using the TransScrip@ One-Step RT-PCR SuperMix (AT411-02, Transgene) according to the manufacturer's instructions.

## RNA sequencing assay

TAMs for RNA sequencing were isolated from MC38-derived tumors inoculated in *Id1^f/f* or *Id1^Lyz-KO* mice as described above. Tumor cells for RNA sequencing were isolated from MC38 cell-derived tumors co-inoculated with *Id1^f/f* or *Id1^Lyz-KO* TAMs. Briefly, tumor cells were isolated sequentially using CD45 negative magnetic sorting (via PE anti-mouse CD45 recombinant antibody (157604, Biolegend), anti-PE microbeads (130-048-801, Miltenyi Biotec)), and Epcam positive magnetic sorting (via anti-Epcam microbeads (130-105-958, Miltenyi Biotec)). Total RNA was extracted using the Trizol reagent according to the manufacturer's protocol. RNA purity and quantification were evaluated using the NanoDrop 2000 spectrophotometer (Thermo-Fisher Scientific). RNA integrity was assessed using the Agilent 2100 Bioanalyzer (Agilent Technologies). Then the libraries were constructed using VAHTS Universal V6 RNA-seq Library Prep Kit according to the manufacturer's instructions. The transcriptome sequencing and analysis were conducted by OE Biotech Co., Ltd.

## Immunoprecipitation, immunoblotting, immunostaining and immunohistochemistry

Co-immunoprecipitation experiment was performed as described previously[52]. Briefly, cells were collected and lysed on ice for 30 min. The cell lysates were incubated with indicated antibodies at 4 °C overnight, followed by incubating with Protein A/G Plus-Agarose (sc-2003, Santa Cruz) at 4 °C for another 2 h. The immunocomplex was washed 4–6 times and boiled in 2 × SDS sample buffer for 5 min. The co-precipitates were resolved using SDS-PAGE and immunoblotted with specific antibodies. Western blot images were captured by the Tanon 5200 chemiluminescent imaging system. Uncropped blots have been provided in the Source Data file and in Supplementary Figs. 10 and 11.

For the multiple immunohistochemistry (mIHC), the formalin-fixed, paraffin-embedded (FFPE) sections were stained using a four-color labeled kit (0001100020, PANOVUE Biotechnology Co., LTD) according to the manufacturer's protocol as described previously[53]. For immunofluorescence staining, cells seeded on coverslips were briefly washed with phosphate-buffered saline (PBS) and fixed with 4% buffered paraformaldehyde for 15 min, permeabilized with 0.5% Triton X-100 for 15 min, blocked with 3% BSA for 30 min at 37 °C, and stained with specific primary antibodies followed by corresponding secondary antibodies. Nuclei were counterstained with DAPI (ZLI-9557, ZSGB-BIO). Immunofluorescent images were captured with a confocal microscope (Olympus, FV3000).

For immunohistochemistry analysis, the paraffin-embedded tissue sections were deparaffinized with xylene and hydrated through graded alcohols into water. Antigen retrieval was carried out with a citrate buffer (10 mM sodium citrate buffer, pH 6.0) at sub-boiling temperature for 15 min. The sections were permeabilized with 0.5% Triton-X100/PBS for 20 min. Endogenous peroxidase activity was blocked with 3% $H_2O_2$ solution for 10 min, followed by washing three times with PBS. Blocking buffer (3% BSA/PBS) was added to the sections and incubated for 30 min. Slides were then incubated with indicated primary antibodies at 4 °C overnight. After washing three times, sections were incubated for 30 min with corresponding secondary antibodies at room temperature. Signals were detected with freshly made DAB substrate solution. Sections were then counterstained with hematoxylin, dehydrated, and mounted with coverslips. Images were captured using an Olympus BX51 microscope and analyzed by Image-Pro Plus 6.0 (Media Cybernetics Inc.).

## Mass spectrometry

For protein identification by mass spectrometry, cell lysates of RAW264.7 cells were collected and immunoprecipitated with anti-Id1 antibody. The eluents were subjected to the EASY-nLC 1000 system which was directly interfaced with the Thermo Orbitrap Fusion mass

spectrometer. Data acquisition was performed by QLBio Biotechnology Co., Ltd and analyzed using an in-house Proteome Discoverer (Version PD1.4, Thermo-Fisher Scientific). Peptides only assigned to a given protein group were considered as unique.

## Invasion assay

Transwell invasion assays were performed using transwell chambers with filter membranes of 8-μm pore size. Chambers were precoated with 10 μg/mL fibronectin (354008, Corning) on the lower surface, and the polycarbonate filter was coated with matrigel (30 μL per well). Then, the chambers were inserted in 24-well culture plates. Single-cell suspension was seeded into the upper chamber ($8 \times 10^4$ cells per well in RPMI-1640 containing 0.4% FBS) with different groups of TAMs seeded in the lower compartment (RPMI-1640 containing 10% FBS). After 24 h, non-invaded cells on the upper side of the filter were removed with a cotton swab. Cells invaded were fixed with 4% paraformaldehyde in PBS, stained with crystal violet, and counted using brightfield microscopy[54].

## Proximity ligation assay

Cells on coverslips were fixed in 4% paraformaldehyde solution for 20 min, permeabilized with 0.5% Triton-X100 for another 20 min, and blocked with Duolink® blocking solution (DUO92008, Merck). Cells were then probed with indicated primary antibodies and treated with the Duolink PLA probes using the Duolink® In Situ Red Starter Mouse/Rabbit kit according to the manufacturer's instructions. Images were captured using a confocal fluorescent microscope (Olympus, FV3000)[55].

## Tumor stem cell frequency assay

Immunodeficient BALB/c nude mice were used in the tumor inoculation experiments. For limiting dilution assay, 100, 300, 1000, 3000, and 10,000 tumor cells were injected subcutaneously into the right flanks of the BALB/c nude mice. The frequency of tumor-initiating cells was calculated using Extreme Limiting Dilution Analysis (ELDA) online software (http://bioinf.wehi.edu.au/software/elda/) 40 days after inoculation.

## Tumor sphere-forming assay

Digested tumor cells were gently prepared into single-cell suspension with the StemXVivo Serum-Free Tumorsphere Media (CCM012, R&D) and plated in 96-well ultralow-attachment plates at 500 cells per well. 30% CM from TAMs were added to the semisolid medium. Images were captured and tumor spheres were counted with an Olympus CKX41 microscope.

## Macrophage polarization

Bone-marrow-derived macrophages, RAW264.7 cells or THP-1 cells were polarized with recombinant murine IFN-γ (315-05, Peprotech) or recombinant human IFN-γ (300-02, Peprotech) at the concentration of 20 ng/mL, together with LPS (100 ng/mL, 916374, Merck) for 24 h to induce pro-inflammatory (M1-like) phenotype. BMDMs were treated with IL-4 (20 ng/mL, 214-14, Peprotech) for 48 h to induce polarization to TAMs.

## T-cell migration assay

For the mouse T cell migration assay, the mouse spleen was homogenized, and single cells were harvested by centrifugation. Red blood cells were lysed by re-suspending the pellet in 2 mL red blood cell lysis buffer and incubated at room temperature for 2 min. The splenocytes were centrifuged, washed, and resuspended at $2 \times 10^6$/mL in RPMI-1640 culture medium containing 10% FBS, 10 ng/mL murine recombinant IL-2 (212-12, Peprotech), 50 μM 2-mercaptoethanol (M8210, Solarbio) and anti-CD28 antibody (10312-25, BioGems). The culture dishes were pre-coated with mouse anti-CD3 antibody (05112-25,

BioGems), and the cells were cultured for 3–5 days. Splenic T cells were then sorted by the CD8+ T cell isolation kit (130-096-495, Miltenyi Biotec) and stained with anti-mouse CD8α APC-Cy7 antibody (100714, Biolegend) for the next co-culture assay.

For the human T cell migration assay, human peripheral blood mononuclear cells (PBMCs) were isolated using Ficoll buffer and cultured in RPMI-1640 culture medium containing 10% FBS, 1% antibiotics, and 10 ng/mL human IL-2. The culture dishes were pre-coated with human anti-CD3 and anti-CD28 antibodies, and the cells were cultured for 5–7 days. Then the CD8+ T cells were sorted using a CD8+ T cells isolation kit (Miltenyi Biotec) and stained with anti-human CD8α APC-Cy7 antibody (0.5 μg/$1 \times 10^6$ cells) for 15 min at room temperature (RT).

The stained CD8+ T cells were adjusted to $2 \times 10^6$ cells/mL. 100 μL of the CD8+ T cells were added into the transwell insert (5.0 μm pore size) to allow their contact with the supernatant culture medium of TAMs seeded in the lower compartment. After 1.5 h of co-incubation, the insert was carefully removed. The migrated CD8+ T cells were collected from the lower compartment and enumerated by flow cytometry with counting beads (424902, Biolegend).

## Carboxyfluorescein succinimidyl ester (CFSE) assay

For mouse T cell proliferation assays, CD8+ T cells were isolated as described above and covalently labeled with CFSE (423801, Biolegend) according to the manufacturer's constructions. The labeled T cells ($2 \times 10^5$ cells/well) were added into the upper transwell insert (0.4 μm pore size) to allow their contact with the supernatant culture medium of different groups of RAW264.7 or TAMs ($1 \times 10^5$ cells/well) seeded in the lower compartment. After 24 h co-culture in the complete cell culture media, the CFSE signal was analyzed by flow cytometry.

## OT1 T cells killing assay

Splenocytes were isolated from OT-1 C57BL/6-Tg (TcraTcrb) 1100Mjb/J. The cells were pelleted, washed, and suspended at $2 \times 10^6$ cells/mL in RPMI-1640 culture medium containing 10% FBS, 10 ng/mL murine recombinant IL-2, and 50 μmol/L 2-mercaptoethanol and 5 μg/mL Ovalbumin 257-264 peptides (S7951, Merck). Splenic OT-1 cells were magnetically purified by CD8+ T Cell Isolation Kit (130-104-075, Miltenyi Biotec). Ova-expressing MC38 cells were cocultured with OT-1-T cells at the ratio of 1:8 (tumor cells: T cells) for 24 h, in which the T cells were pre-cocultured with *Id1^{f/f}* or *Id1^{Lyz-KO}* TAMs for 24 h. All cells were collected by trypsinization and stained with PI (P8080, Solarbio) for 15 min. Tumor cell apoptosis was determined by flow cytometry analysis.

## ALDH/CD44 profiling by flow cytometry

Tumor cells were co-cultured with TAMs seeded in the upper insert of the transwell chamber (0.4 μm) for 48 h and were then digested by trypsin. The ALDEFLUOR kit was used to detect the ALDH1 enzymatic activity of CRC cells. Briefly, $5 \times 10^5$/mL cells were suspended in an ALDEFLUOR assay buffer. For each sample, cell aliquots were incubated with BODIPY-aminoacetaldehyde (BAAA) and with or without 75 μM diethylaminobenzaldehyde (DEAB), an ALDH-specific inhibitor at 37 °C for 40 min. Tumor cells were then stained with APC anti-mouse/human CD44 antibody (103011, Biolegend) for flow cytometry analysis. ALDEFLUOR staining was detected using the FITC channel, and CD44 staining was detected using the APC channel of FACs. The fluorescence intensity of the stained cells was collected by using a BD FACSVerse Flow cytometer.

## Statistics and reproducibility

Analyses were performed using GraphPad Prism 8.0 software with significance set to $P < 0.05$. The Kaplan–Meier method was used for survival estimation, and the log-rank test was used for comparisons. The number of samples and independent biological experimental repeats were indicated in the figures or figure legends. Comparisons between two unpaired groups were

evaluated by parametric Student's $t$-tests, Welch's tests, or non-parametric Mann–Whitney $U$ tests. Variance between groups was tested by one-way analysis of variance (ANOVA) if the data were normally distributed. Variance between groups was tested by a Kruskal–Wallis tests if the data were not normally distributed. Two-sided tests are used in the statistics. A Shapiro–Wilk test was used to test the data normality. A Levene test was used to test the variance homogeneity. All quantitative data are expressed as means ± SEM. All blots, and images of IHC or IF are representative and each staining was repeated for at least three biological replicates showing similar results.

### Reporting summary

Further information on research design is available in the Nature Portfolio Reporting Summary linked to this article.

## Data availability

The raw and processed RNA-seq data used in this study are available in the GEO database under accession numbers GSE188572 and GSE200855. The mass spectrometry proteomics data used in this study are available in the ProteomeXchange Consortium (iProX) under accession number PXD046173 [https://www.iprox.cn//page/project.html?id=IPX0007320000]. The publicly available RNA-seq data used in this study are available in the GEO database under accession number GSE80065. The remaining data are available within the Article, Supplementary Information, or Source Data file. Source data are provided with this paper.

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

## Acknowledgements

This work was supported by grants from the National Key Research and Development Program of China (2022YFC2504002, 2022YFC2504003), National Natural Science Foundation of China (81973344 and 82273947 to F.H.; 81903636 to S.S.), the Beijing Natural Science Foundation (7232257 to F.H.). CAMS Innovation Fund for Medical Sciences (CIFMS) (2021-I2M-1-021 to F.H. and S.S., and 2022-I2M-JB-011 to F.H.), Chinese Academy of Medical Sciences Central Public-interest Scientific Institution Basal Research Fund (2018PT35004) and Beijing Outstanding Young Scientist Program (BJJWZYJH01201910023028). We appreciate Beijing Qinglian Biotech Co., Ltd. for the mass spectrometric analysis. Elements of Figs. 1g, 2a, e, l, p, 3j, 4c, g, q, 5a, 7a, e, j, l, s, w and Supplementary Figs. 1a, e, i, m, q, 2a, e, 7a, i, p are created with BioRender.com. Elements of Supplementary Figs. 5h and 8 are reproduced with permission from Springer Nature[56].

## Author contributions

F.H. conceptually planned and supervised the study. F.H. and S.S. designed experiments. S.S. and C.Y. performed most of the experiments unless specified. F.H., S.S., C.Y., and F.C. analyzed data. F.C., J.L., X.-X.L., and C.Z. participated in the animal studies. H. Z., S.-Y. D., X.-T.L, J.-W.S., and J.-J.Y. participated in the molecular and cellular biological experiments. J.-C.Z., X.-W.Z., P.-P.L., and B.C. helped the pathology-related studies. F.H., S.S., and C.Y. wrote the manuscript. S.T.Z. participated in molecular and cellular biological experiments. R.-S.X., Q.Z., and S.-B.L. participated in the clinical sample and information collection. H.Z.Z. conceptualized clinical aspects and contributed to clinical sample collection and data analysis. All authors have read and approved the manuscript.

## Competing interests

The authors declare no competing interests.
