## [Peer Review File · Nature Communications]

ID1 Expressing Macrophages Support Cancer Cell Stemness and Limit CD8+ T Cell Infiltration in Colorectal CancerREVIEWER COMMENTS

Reviewer #1 (Remarks to the Author): with expertise in colorectal cancer, immunology, stem cells

Summary: Shang et al. present an interesting study showing that ID1 expression in tumor associated macrophages (TAMs) plays a critical role in promoting tumor growth. The authors suggest that the mechanism for tumor growth and metastasis is by reducing CD8+ T cell recruitment to the tumor, and promotion of cancer stem cells within the tumor. The authors provide several lines of experiment data showing that ID1 knockout in TAMs is associated with reduced tumor growth when compared to standard TAMs (having normal ID1). There are strong data provided to show that increased ID1 expression in TAMs leads to increased tumor growth and conversely, that ID1 knockout in TAMs leads to reduced tumor growth and metastasis in association with reduced T cell infiltration to the tumor, as well as, increased CD44+ cancer stem cells.

Overall, this paper presents an interesting set of data and conclusions. The strengths are there are lots of data provided using immunodeficient and immunocompetent mouse models in which similar effects of ID1 are seen. Data are provided using ID1 KO TAMs and ID1 overexpressing RAW macrophages. The major limitations of the study, however, are as follows:

- 1) All the data provided is limited to mouse cancer cell lines (CT-26 and MC38); The relevance to human tumors is limited in that there are no data provided examining patient derived tumors. Thus, it is not clear as to whether ID1's role in human TAMs is indeed the same with human colorectal tumors as it is with mouse tumors. It would have been nice to see at least some validation using human cancer lines even in vitro (i.e. co-culture with macrophages)
- 2) All data provided is limited to mouse xenograft tumors and 2 mouse CRC cell lines. Are the results shown only relevant to CRC or do the same principles of ID1 expression in TAMs also apply to other tumor types as well? In other words, can ID1 inhibition be used to increase T cells with other tumors as well or is this only true of CRC and if so why?
- 3) The link between ID1 in macrophages and colorectal cancer stem cells is weak. Although there is clear association with CD44+ cell number in the tumors with increased ID1. The cancer stem cell capacity data is not sufficient to draw a strong link to ID1 expression. In

particular, there is no mechanistic link to cancer stem cells provided and it is not obvious as to how ID1 expression in macrophages leads to increased CSC number. If the authors want to make this a major conclusion including in the title, then it would be nice to see some justification as how increased CSCs are formed as a result of ID1 expression.

Reviewer #2 (Remarks to the Author): with expertise in cancer immunology

I find this a well written and carefully performed study which demonstrates the biological significance of ID1 in TAM in relation to regulating tumor growth. It points at the possibility of targeting ID1 and thereby change the phenotype of TAM from being immunosuppressive to proinflammatory with the ability to activate T cells. The strength of the study is that it also includes several mouse models showing the effect of delaying tumor growth and metastasis. Data are well presented in a clear and well written language and their conclusions are well supported by their presented results. Their results add significantly to the knowledge of ID1 regulation of myeloid cells and on TAM, where relatively little is known from the literature.

I miss a better discussion on how their findings focusing on TAM relates to previous publications by others where the focus has been on MDSCs and DCs. Although they refer to one MDSC study in their discussion (ref 22), they have made no effort in characterizing ID1 expression in MDSCs or isolating them in their models here. DCs are not even mentioned, and they are key to activating CD8+ T cells. There is no discussion on the possible relationship between their immune suppressive ID1 TAM and MDSCs

They elegantly demonstrate the effect of TAM on CD8+ cells in their adoptive transfer model with Id1Lyz-KO TAMs. When they study effects of pharmacologic elimination of ID1 expression by ML323, the role of TAM suppression is not as conclusively shown. Although they can demonstrate that reduction of ID1 expression in TAM by ML323 administration results in a marked tumor inhibition, the mechanism behind this is only indirectly inferred to by increased tumor-infiltrating CD8+ cells and enhanced expression of Granzyme B and IFN γ expression. Conclusive experiments demonstrating e.g. direct functional effects on CD8+

cells of isolated TAM, from ML 323 treated or untreated tumors, would be the most convincing way to show this.

Reviewer #3 (Remarks to the Author): with expertise in colorectal cancer

Cancer therapy is facing challenges in eliminating CSCs and improving anti-tumor immunity. TAMs are the main immune cell population in tumor tissues and contribute to the formation of the CSC niche and suppressive immune microenvironment. This study reveals that high expression of ID1 in TAMs is associated with poor outcomes in CRC patients. ID1-expressing macrophages maintain cancer stemness and impede CD8+ T cell infiltration. ID1 interacts with STAT1 to induce its cytoplasmic distribution, which inhibits STAT1-mediated transcription of SerpinB2 and CCL4. These two secretory factors are responsible for the inhibition of cancer stemness and the recruitment of CD8+ T cells. Pharmacologically reducing ID1 expression can ameliorate CRC progression and enhance the sensitivity to immunotherapy and chemotherapy. In summary, this study highlights the crucial role of ID1 in controlling the pro-tumor phenotype of TAMs and provides a basis for therapeutic intervention of ID1 in CRC therapy.

This article is highly innovative, well-written, and the research design is quite rigorous. However, there are still some shortcomings, including the following:

In Figure 1, since c-Myc is the most significantly up-regulated oncogene selected in this research system, it is recommended to improve the understanding of TAMs expressing c-Myc and their role in promoting the proliferation and metastasis of CRC.

In Figure 1c, it is suggested to verify the protein abundance of the other top 4 genes, in addition to ID1 in MC38-derived CM treated RAW 264.7 cells.

In Extended Data Fig. 1d, 1h, 1i, 1p, the tumor diameter exceeding 2cm in some cases, does it violate the relevant regulations of animal welfare?

In Figure 2e, in the mouse model of colon cancer liver metastasis, the orthotopic tumor liver

metastasis model can better simulate the situation of colon cancer liver metastasis. If possible, it is suggested to improve the experiment in this aspect.; In addition, lung metastasis is also a common metastatic way in colon cancer, and the tail vein injection of tumor cells in mice can be adopted to suggest lung metastasis models for related experiments.

In Figure 4, it is suggested to analyze CRC stemness by staining for ALDH, CD133 simultaneously, reported markers of CRC stem cells, not only CD44.

REVIEWER COMMENTS

Reviewer #1 (Remarks to the Author): with expertise in colorectal cancer, immunology, stem cells

Summary: Shang et al. present an interesting study showing that ID1 expression in tumor associated macrophages (TAMs) plays a critical role in promoting tumor growth. The authors suggest that the mechanism for tumor growth and metastasis is by reducing CD8⁺ T cell recruitment to the tumor, and promotion of cancer stem cells within the tumor. The authors provide several lines of experiment data showing that ID1 knockout in TAMs is associated with reduced tumor growth when compared to standard TAMs (having normal ID1). There are strong data provided to show that increased ID1 expression in TAMs leads to increased tumor growth and conversely, that ID1 knockout in TAMs leads to reduced tumor growth and metastasis in association with reduced T cell infiltration to the tumor, as well as, increased CD44⁺ cancer stem cells.

Overall, this paper presents an interesting set of data and conclusions. The strengths are there are lots of data provided using immunodeficient and immunocompetent mouse models in which similar effects of ID1 are seen. Data are provided using ID1 KO TAMs and ID1 overexpressing RAW macrophages. The major limitations of the study, however, are as follows:

- 1) All the data provided is limited to mouse cancer cell lines (CT-26 and MC38); The relevance to human tumors is limited in that there are no data provided examining patient derived tumors. Thus, it is not clear as to whether ID1's role in human TAMs is indeed the same with human colorectal tumors as it is with mouse tumors. It would have been nice to see at least some validation using human cancer lines even in vitro (i.e. co-culture with macrophages)

Re: Thanks for your comments. According to your suggestions, a set of experiments were performed to confirm whether ID1 in human TAMs plays the same role as we observed in mouse tumors. (1) ID1 was depleted in primary TAMs isolated from human CRC tissues with *ID1-shRNA* expressing lentivirus. Primary TAMs with *ID1* depletion or not were cocultured with T cells (isolated from CRC patients' peripheral blood mononuclear cell) or HCT116 cells to observe how ID1 expressing human primary TAMs affected T cell migration and cancer cell stemness features. We found that depletion of ID1 in primary TAMs promoted the migration ability of CD8⁺ T cells, decreased the stemness traits of tumor cells. (2) We also observed that depletion of ID1 in human TAMs increased the protein expression of SerpinB2 and CCL4. These data were presented as revised Fig. 3q, 4d, 4f, 5e and 5i.

Besides, in the previous version, we have tested the effects of ID1 overexpression in M1-like human THP-1 cells (considered as M1-like macrophages) on the stemness traits of different human colon cancer cell lines, such as DLD-1, HCT-8 and HCT116 (presented as revised Fig. 4s and Extended Data Fig. 3c, f and h).

- 2) All data provided is limited to mouse xenograft tumors and 2 mouse CRC cell lines. Are the results shown only relevant to CRC or do the same principles of ID1 expression in TAMs also apply to other tumor types as well? In other words, can ID1 inhibition be used to increase T cells with other tumors as well or is this only true of CRC and if so why?

Re: Thanks a lot for your constructive comments! We have expanded our study to liver cancer and pancreatic cancer. Conditional medium (CM) from *Id1^{fl/fl}* or *Id1^{Lyz-KO}* BMDM-derived TAMs were intratumorally injected in H22 (mouse hepatocellular carcinoma cell line) and Pan02 (mouse pancreatic ductal adenocarcinoma cell line) tumor-bearing mice. Our data indicated that deletion of *Id1* in TAMs also alleviated tumor growth, increased CD8⁺ T cell infiltration and activation, and reduced tumor cell stemness. These data validated the generalizability of ID1 expressing TAMs in the induction of cancer immune evasion and in the maintenance of cancer stemness traits. These new data were presented as Extended Data Fig. 7i-v in the revised manuscript.

3) The link between ID1 in macrophages and colorectal cancer stem cells is weak. Although there is clear association with CD44⁺ cell number in the tumors with increased ID1. The cancer stem cell capacity data is not sufficient to draw a strong link to ID1 expression. In particular, there is no mechanistic link to cancer stem cells provided and it is not obvious as to how ID1 expression in macrophages leads to increased CSC number. If the authors want to make this a major conclusion including in the title, then it would be nice to see some justification as how increased CSCs are formed as a result of ID1 expression.

Re: Thanks so much for your constructive suggestion. In our previous study, we found that ID1 inhibits the transcription and secretion of *Serp1B2* in TAMs. Besides, ID1 enhances FAK activation, a critical signaling in maintaining cancer stemness traits (*Exp Mol Med.* 2020. PMID: 32514188). *Serp1B2* is known to inhibit urokinase (uPA), decreases uPAR activation and downregulates the downstream FAK-signaling (*Nat Rev Mol Cell Biol.* 2010. PMID: 20027185). Based on the evidence, we proposed that *Serp1B2* (TAMs)—FAK (CRCs) regulatory axis acts as the bridge to connect cancer stemness with high ID1 expression in TAMs. However, the molecular mechanistic link between FAK and CRC stemness traits still needs to be strengthened. Therefore, we performed additional experiments from two aspects: 1) We deciphered which critical stemness related signaling pathway mediated the stemness maintaining capacity of ID1 in TAMs. 2) We isolated primary TAMs from human CRC tissues to establish the clinical relevance of TAMs-derived ID1 in cancer stemness maintaining. The details are as follows:

- 1) ① GSEA analysis suggested that depletion of *Id1* in TAMs downregulates YAP, but not Notch, Wnt/ β -catenin and SHH signaling pathways (widely accepted tumor stemness related signaling pathways) in CRCs (Fig. 4h and i; Extended Data Fig. 3i). ② YAP is reported as a classical downstream target protein of FAK signaling. FAK induced YAP^{Y357} phosphorylation is known to enhance YAP protein stability, nuclear translocation and transcriptional activation (PMID: 34052254). We did find that high ID1 expression in TAMs connected with FAK activation, upregulated YAP^{Y357} phosphorylation, enhanced YAP protein stability, nuclear translocation and subsequent activated YAP-TEAD transcription (Fig. 4j-n; Extended Data Fig. 3j-q). Y15, a specific inhibitor of FAK, could reverse the upregulation role of *ID1* overexpression in M1-like THP1 cells on YAP protein expression, indicating that the ID1 induces YAP activation via a FAK-dependent manner (Fig. 4k). Moreover, verteporfin, a suppressor of YAP-TEAD transcriptional complex, largely abrogated the CRC tumor stemness promoting effect of CM from M1-like THP-1 cells with *ID1^{OE}*, suggesting that the tumor stemness maintaining role of ID1 in TAMs is mediated by YAP (Fig. 4p).
- 2) ① HCT116 cells cocultured with CRC patients tumor tissues derived *ID1^{KD}* TAMs expressed lower levels of CD44 and LGR5 than those cocultured with *Ctrl* TAMs (Fig. 4d). ② CM from human *ID1^{KD}* TAMs inhibited the tumor sphere formation ability (Fig. 4f). These data, together with our previous findings validated the tumor stemness maintaining role of ID1 in TAMs from two aspects: CRC mouse models and clinical CRC tumor samples.

Therefore, we conclude that TAMs-derived ID1 inhibits the transcription of *Serp1B2* to induce FAK-YAP axis activation, thus enhancing colorectal cancer stemness traits.

Reviewer #2 (Remarks to the Author): with expertise in cancer immunology

I find this a well written and carefully performed study which demonstrates the biological significance of ID1 in TAM in relation to regulating tumor growth. It points at the possibility of targeting ID1 and thereby change the phenotype of TAM from being immunosuppressive to proinflammatory with the ability to activate T cells. The strength of the study is that it also includes several mouse models showing the effect of delaying tumor growth and metastasis. Data are well presented in a clear and well written language and their conclusions are well supported by their presented results. Their results add significantly to the knowledge of ID1 regulation of myeloid cells and on TAM, where relatively little is known from the literature.

- 1) I miss a better discussion on how their findings focusing on TAM relates to previous publications by others where the focus has been on MDSCs and DCs. Although they refer to one MDSC study in their discussion (ref 22), they have made no effort in characterizing ID1 expression in MDSCs or isolating them in their models here. DCs are not even mentioned, and they are key to activating CD8⁺ T cells. There is no discussion on the possible relationship between their immune suppressive ID1 TAM and MDSCs.

Re: Thanks for your constructive comments! It's indeed very important to discuss the relationship about the immune suppressive role of ID1 between TAMs and MDSCs. ID1 was reported to promote the switch from DCs differentiation to MDSCs expansion during tumor progression, which inhibits effective T cells proliferation and activation (Ref. 22, PMID: 25924227). In this article, ID1 has a quite low expression either in naïve DC or DCs from tumor bearing mice compared to MDSCs. Therefore, we wondered whether manipulation of ID1 in MDSCs also influence CD8⁺ T cell recruitment. CD11b⁺ Gr1⁺ MDSCs were isolated from s.c. MC38 tumors in *Id1^{ff}* and *Id1^{Lyz-KO}* mice, and cocultured with CD8⁺ T cells. We found that depletion of *Id1* in MDSCs had no effect on CD8⁺ T cells migration (Extended Data Fig. 2m in the revised manuscript). These data suggested that ID1 expressing TAMs but not MDSCs hamper CD8⁺ T cell recruitment and promote the evasion of tumor cells from immune elimination. We also discussed the different immune suppressive role of ID1 in regulating CD8⁺ T cells between TAMs and MDSCs in the revised MS.

- 2) They elegantly demonstrate the effect of TAM on CD8⁺ cells in their adoptive transfer model with *Id1^{Lyz-KO}* TAMs. When they study effects of pharmacologic elimination of ID1 expression by ML323, the role of TAM suppression is not as conclusively shown. Although they can demonstrate that reduction of ID1 expression in TAM by ML323 administration results in a marked tumor inhibition, the mechanism behind this is only indirectly inferred to be by increased tumor-infiltrating CD8⁺ cells and enhanced expression of Granzyme B and IFN γ expression. Conclusive experiments demonstrating e.g. direct functional effects on CD8⁺ cells of isolated TAM, from ML 323 treated or untreated tumors, would be the most convincing way to show this.

Re: Thanks for your constructive suggestions! We performed two sets of experiments to address this issue. (1) TAMs of s.c. CT26 tumors in BALB/c mice treated with or without ML323 (5 mg/kg) for 20 days were isolated using magnetic cell sorting. The effects of TAMs on CD8⁺ T cell migration, CRC tumor cell stemness, as well as the protein expression of Serpinb2 and Ccl4 in isolated TAMs were tested *in vitro*. TAMs isolated from tumors treated with ML323 upregulate the secretion of Serpinb2 and Ccl4, promote CD8⁺ T cell migration and decrease the CD44^{high} Aldh⁺ cell ratio in CT26 cells compared to the Vehicle group (Fig. 7l-n; the Extended Data 6k, l in the revised manuscript). (2) We evaluated the tumor inhibition role of ML323 in *Id1^{ff}* and *Id1^{Lyz-KO}* mice bearing s.c. MC38 tumors. ML323 administration demonstrated a more pronounced tumor inhibitory effect in *Id1^{ff}* mice than that in *Id1^{Lyz-KO}* mice (Fig. 7j, k and Extended Data Fig. 6h in the revised manuscript). These data confirmed that ML323 treatment leads to the recruitment of CD8⁺ T cells and inhibition of CRC stemness, which is attributed to the alleviation of TAMs' tumor-promoting effect by reducing *Id1* expression.

Reviewer #3 (Remarks to the Author): with expertise in colorectal cancer

Cancer therapy is facing challenges in eliminating CSCs and improving anti-tumor immunity. TAMs are the main immune cell population in tumor tissues and contribute to the formation of the CSC niche and suppressive immune microenvironment. This study reveals that high expression of ID1 in TAMs is associated with poor outcomes in CRC patients. ID1-expressing macrophages maintain cancer stemness and impede CD8⁺ T cell infiltration. ID1 interacts with STAT1 to induce its cytoplasmic distribution, which inhibits STAT1-mediated transcription of SerpinB2 and CCL4. These two secretory factors are responsible for the inhibition of cancer stemness and the recruitment of CD8⁺ T cells. Pharmacologically reducing ID1 expression can ameliorate CRC progression and enhance the sensitivity to immunotherapy and chemotherapy. In summary, this study highlights the crucial role of ID1 in controlling the pro-tumor phenotype of TAMs and provides a basis for therapeutic intervention of ID1 in CRC therapy.

This article is highly innovative, well-written, and the research design is quite rigorous. However, there are still some shortcomings, including the following:

- 1) In Figure 1, since c-Myc is the most significantly up-regulated oncogene selected in this research system, it is recommended to improve the understanding of TAMs expressing c-Myc and their role in promoting the proliferation and metastasis of CRC.

Re: Thank you for your valuable comments! We have thoroughly reviewed the related studies. It has been observed that c-Myc is significantly expressed in TAMs of neuroblastoma, lung cancer, and colorectal cancer (Blood 2012, PMID: 22067385; Oncotarget 2017, PMID: 29207662; Aging 2020, PMID: 33260160). Furthermore, these studies reported that c-Myc plays a role in promoting M2 polarization of macrophages by activating SCARB1, ALOX15, MRC1, and other factors. However, the precise molecular mechanisms through which c-Myc-expressing TAMs promote tumor progression remain to be comprehensively understood. This aspect holds potential for exploration, and we are dedicated to pursuing further research in this field.

- 2) In Figure 1c, it is suggested to verify the protein abundance of the other top 4 genes, in addition to ID1 in MC38-derived CM treated RAW 264.7 cells.

Re: Thanks for your comments! We have verified the protein abundance of c-Myc, Tcf4, Id2 and Snai1 in MC38-CM treated RAW 264.7 cells and got the same conclusion as in CT26-CM treated ones. The new data was presented as Fig. 1c in the revised manuscript.

- 3) In Extended Data Fig. 1d, 1h, 1i, 1p, the tumor diameter exceeding 2 cm in some cases, does it violate the relevant regulations of animal welfare?

Re: Thanks so much for your criticism! Due to the unskilled inoculation technique of the operator, some subcutaneous tumors form two tumor masses. The individual diameter of each tumor did not exceed 2 cm, however the combined diameter did exceed 2 cm in Extended Data Fig. 1d, h and p. With the strict attitude toward science, we repeated these experiments and made replacement of the relevant data in the revised manuscript.

- 4) In Figure 2e, in the mouse model of colon cancer liver metastasis, the orthotopic tumor liver metastasis model can better simulate the situation of colon cancer liver metastasis. If possible, it is suggested to improve the experiment in this aspect.; In addition, lung metastasis is also a common metastatic way in colon cancer, and the tail vein injection of tumor cells in mice can be adopted to suggest lung metastasis models for related experiments.

Re: Thanks so much for your instructive comments! The orthotopic colorectal cancer liver metastasis model and the experimental lung metastasis model were used to confirm the tumor metastasis promoting role of Id1 in TAMs. Conditional medium (CM) from *Id1^{fl/fl}* and *Id1^{Lyz-KO}* TAMs were administrated via intraperitoneal injection every other day after tumor injection. Less metastases in the liver and in the lung were identified in the *Id1^{Lyz-KO}* group in comparison with the *Id1^{fl/fl}* group (Fig. 2l-o and Extended Data Fig. 1q-s in the revised manuscript). Meanwhile, increased CD8⁺ T cells infiltration and decreased percentage of CD44^{high} Lgr5⁺ tumor stem cells were observed in the *Id1^{Lyz-KO}* group in comparison with the *Id1^{fl/fl}* group (Fig. 3g, h; Fig. 4b and Extended Fig. 3a in the revised manuscript). These data further supported our conclusion that ID1-expressing TAMs play critical role in promoting CRC metastasis.

- 5) In Figure 4, it is suggested to analyze CRC stemness by staining for ALDH, CD133 simultaneously, reported markers of CRC stem cells, not only CD44.

Re: Thanks for your comments! We re-analyzed CRC stemness with a strategy by detecting two stem cell markers together. We chose two stem cell markers LGR5 and ALDH in combination with CD44. LGR5 is a widely recognized indicator of colorectal cancer stemness. Its detection is convenient and can be effectively performed even with a small number of cells (PMID: 29477614, 18808327). ALDH enzymatic activity display characteristics of cancer stem cells, which can be measured with Aldefluor flow cytometry-based assay. However, larger cell number is needed for Aldefluor assay than LGR5 staining assay. Therefore, Aldefluor assay was used in the in vitro systems, and LGR5 staining was performed for the in vivo systems and the primary TAMs related assays (the cell number is relatively low).

Aldefluor assay together with CD44 staining data were presented in the revised Fig. 4e, 7n and Extended Data Fig. 3b, 3c. LGR5 and CD44 simultaneous staining data were presented in the revised Fig. 4a, 4b, 4d and Extended Data Fig. 3a, 7h and 7v. These data confirmed that depletion of Id1 in TAMs decreased CD44^{high}ALDH⁺ or CD44^{high}LGR5⁺ tumor stem cells ratio in vitro and in vivo.

REVIEWERS' COMMENTS

Reviewer #1 (Remarks to the Author):

In the reviewing the revised manuscript, the authors have done a great job at addressing my concerns. In particular in the inclusion of several pieces of data using human cell lines and even primary human macrophage data with ID1 knockdown/knockout greatly strengthens the manuscript. These data provide credence to their conclusions that ID1 in macrophages plays an important role in the colorectal cancer microenvironment.

The new data on cancer stem cells is improved although functional characterization of cancer stemness is not still provided. Given the changes made, the manuscript is significantly improved and the data now in accordance to the conclusions made in the paper.

Reviewer #2 (Remarks to the Author):

The authors have responded to the critique from me and the other reviewers in a satisfactory way. Also, the comments from me and the other reviewers has resulted in additional experiments which considerably improves the quality of the manuscript.

Reviewer #3 (Remarks to the Author):

The authors did a good job of answering and addressing the questions posed by the reviewers. This manuscript is recommended to be published.

REVIEWERS' COMMENTS

Reviewer #1 (Remarks to the Author):

In the reviewing the revised manuscript, the authors have done a great job at addressing my concerns. In particular in the inclusion of several pieces of data using human cell lines and even primary human macrophage data with ID1 knockdown/knockout greatly strengthens the manuscript. These data provide credence to their conclusions that ID1 in macrophages plays an important role in the colorectal cancer microenvironment.

The new data on cancer stem cells is improved although functional characterization of cancer stemness is not still provided. Given the changes made, the manuscript is significantly improved and the data now in accordance to the conclusions made in the paper.

Re: Thank you very much for your positive comments. We really appreciate your professional suggestions that has enabled us to make further improvements to our manuscript.

Reviewer #2 (Remarks to the Author):

The authors have responded to the critique from me and the other reviewers in a satisfactory way. Also, the comments from me and the other reviewers has resulted in additional experiments which considerably improves the quality of the manuscript.

Re: Thank you very much for your positive comments. We really appreciate your professional suggestions that has enabled us to make further improvements to our manuscript.

Reviewer #3 (Remarks to the Author):

The authors did a good job of answering and addressing the questions posed by the reviewers. This manuscript is recommended to be published.

Re: Thank you very much for your positive comments. We really appreciate your professional suggestions that has enabled us to make further improvements to our manuscript.